# DualTimesField: Rethinking Time Series as Continuous-Time Trends and Events

Wencheng Zhang [* 1]   Long Li [* 1]   Huayi Qin [* 1]   Zongjuan Wu [1]   Jing Li [1]   Wanghu Chen [† 1]

## Abstract

Effective time series representation is critical for revealing temporal dynamics in many fields. However, existing approaches encounter fundamental limitations. Discrete-time representations struggle with irregular sampling and the trade-off of fidelity and efficiency, while traditional implicit neural representations suffer from spectral bias and frequency entanglement. To address these challenges, we conceptualize time series as the superposition of continuous trends and discrete events from a continuous-time perspective and propose DualTimesField, a framework that utilizes dual implicit neural fields. Its Continuous Time Field captures smooth trends through bandwidth-limited parameterization, while a Discrete Geometric Field models transient events using learnable Gabor atoms, gated sparsity, and coarse-to-fine scale annealing. This explicit field separation effectively overcomes both limitations. Experiments on nine real-world benchmarks demonstrate substantial improvements in representation fidelity, achieving 51.2% average MSE reduction over discrete-time baselines and competitive interpolation on irregular data. Code is available at https://github.com/WisdomTogether/DualTimesField.

## 1. Introduction

Time series data serves as a crucial medium for describing complex dynamic phenomena across various fields (Fulcher et al., 2020). Its effective representation is foundational for uncovering underlying dynamical structures and facilitating downstream analytical tasks (Fons et al., 2022). However, real-world temporal processes exhibit distinct characteristics at different scales. Slowly evolving baseline dynamics often

coexist with abrupt and localized changes, with trends interwoven with events. A central question, therefore, is *how to represent time series* in a way that naturally separates smooth trends from sparse events.

Because physical systems usually operate in continuous time domains, existing methods predominantly discretize time into fixed-step sampling points and model it in discrete index space (Trirat et al., 2024). Such discretization inherently presupposes observations at regular intervals, which imposes intrinsic limitations on representation fidelity (Rubanova et al., 2019). According to the Nyquist–Shannon sampling theorem, distortion-free reconstruction under regular sampling requires the sampling rate to exceed twice the signal bandwidth; otherwise, spectral aliasing results in an irreversible loss of high-frequency information (Shannon, 1949). This creates a fundamental tension between representation fidelity and efficiency, since a higher sampling rate improves faithfulness but incurs prohibitive storage and computational costs, whereas a lower rate risks permanently losing transient events. A more fundamental issue arises when real-world observations depart from regular intervals altogether, as in the case of irregular timestamps or missing values. The discrete index assumption no longer holds, and conventional models must rely on interpolation or imputation that introduces additional error (Deng et al., 2021). Beyond these limitations of regular-interval modeling, discrete representations also overlook intrinsic temporal correlations by treating observations as independent snapshots rather than manifestations of an underlying continuous process. This fundamentally limits the capacity for capturing continuous temporal dynamics and meaningful multi-scale structure (Ni et al., 2022).

Implicit Neural Representations (INRs) address discretization by encoding time series as continuous coordinate-to-value functions. This formulation enables resolution-agnostic querying and supports irregular sampling. However, INRs still face two critical challenges (Essakine et al., 2025). First, spectral bias causes implicit networks to preferentially capture low-frequency components while underfitting high-frequency structures (Fons et al., 2022). This directly degrades representation fidelity for transient events and abrupt changes characterizing real-world time series. Although positional encodings (Tancik et al., 2020) and multi-resolution architectures (Kazerouni et al., 2024) partially address this

[*]Equal contribution   [†]Corresponding author   [1]College of Artificial Intelligence and Computer Science, Northwest Normal University, Lanzhou, Gansu, China. Correspondence to: Wanghu Chen <chenwh@nwnu.edu.cn>.

*Proceedings of the 43rd International Conference on Machine Learning*, Seoul, South Korea. PMLR 306, 2026. Copyright 2026 by the author(s).

issue, they remain constrained by implicit frequency entanglement within shared embedding spaces. Second, unlike traditional signal processing methods that explicitly decompose signals into interpretable frequency bands (Deznabi & Fiterau, 2023), current INR approaches lack structured frequency separation mechanisms (Ko et al., 2025). This causes long-term trends and short-term events to interfere within a single implicit field, precluding physically meaningful decomposition and obscuring the natural separation between smoothly evolving dynamics and abrupt changes.

To address these issues, we rethink the time series representation paradigm itself in this study. Rather than modeling time series as a single implicit field, we reconceptualize them as the superposition of two distinct temporal phenomena and propose DualTimesField with dual implicit neural fields. The Continuous Time Field captures smoothly evolving trends characterizing baseline dynamics, while the Discrete Geometric Field represents localized transient events with explicit spatiotemporal semantics. This decomposition explicitly separates trends from events, alleviating spectral bias and frequency entanglement while yielding high-fidelity and interpretable continuous-time representations. In summary, our contributions are as follows:

- We propose DualTimesField, a dual implicit neural field that separates continuous-time representations into bandwidth-limited trends and sparse events to alleviate spectral bias and frequency entanglement in INRs.

- We introduce a bandwidth-limited trend parameterization with explicit spectral cutoffs to separate the two fields and mitigate spectral entanglement.

- We design an event module with learnable Gabor atoms, gated sparsity, and coarse-to-fine scale annealing to capture transient events.

- We evaluate representation quality on nine long-horizon and six irregular benchmarks, reducing reconstruction MSE by 51.2% over TimesNet and achieving state-of-the-art interpolation on five of six.

## 2. Related Work

We position DualTimesField as a time series representation learning framework that seeks continuous, compact, and queryable encodings. Traditional representation methods employ symbolic discretization techniques (Lin et al., 2007; Ye & Keogh, 2009), trading resolution flexibility for interpretability. Deep learning approaches have significantly advanced time series modeling through adaptive representations. Autoencoders (Malhotra et al., 2016; Franceschi et al., 2019) learn compact encodings, contrastive learning frameworks (Tonekaboni et al., 2021; Yue et al., 2022) capture temporal invariances, while Transformers (Zhou et al., 2021;

Wu et al., 2021) and state space models (Gu et al., 2022; Gu & Dao, 2024) learn long-range dependencies efficiently. Despite their success, these methods operate on discrete sampling grids, limiting their ability to naturally handle irregular observations or support resolution-agnostic queries (Deng et al., 2021). This discrete paradigm motivates the exploration of continuous representations via implicit neural fields, with explicit separation of trends and events as a natural extension for capturing multi-scale temporal structure.

Implicit neural representations (INRs), or neural fields, encode signals through continuous functions parameterized by multi-layer perceptrons (Essakine et al., 2025). While successful in rendering (Mildenhall et al., 2021), applying INRs to time series is challenging. To mitigate spectral bias in standard MLPs, recent methods adopt sinusoidal activations such as SIREN (Sitzmann et al., 2020) or Fourier feature positional encodings (Tancik et al., 2020) to improve high-frequency fitting. Building on these advances, HyperTime (Fons et al., 2022) demonstrates the utility of INRs for time series reconstruction and imputation using periodic activations. For modeling irregular observations, Neural ODEs (Chen et al., 2018) and Latent ODEs (Rubanova et al., 2019) define continuous-time latent dynamics through differential equations, naturally accommodating arbitrary time gaps between observations. However, these approaches model the signal as a single holistic function without explicitly separating frequency components. Even with positional encodings or multi-resolution designs (Tancik et al., 2020; Ko et al., 2025), spectral bias persists, with trend and transient components remaining entangled within shared parameters. This motivates our dual-field architecture, which integrates structured multi-scale separation into continuous neural representations.

To address frequency entanglement in INRs, we draw inspiration from multi-scale decomposition techniques. Classical signal processing provides mathematical tools such as Fourier analysis for global frequency characterization (Osgood, 2007) and wavelet transforms for localized time-frequency representations (Mallat, 1989). Recent deep learning methods incorporate decomposition as learnable inductive bias. N-BEATS (Oreshkin et al., 2020) employs interpretable basis expansions with dedicated stacks for trend and seasonal components, while Autoformer (Wu et al., 2021) and MultiWave (Deznabi & Fiterau, 2023) integrate decomposition mechanisms into neural architectures. Frequency-domain models such as Fourier Neural Operators (Li et al., 2021) operate in the spectral domain but learn holistic mappings rather than explicit component decomposition. Nevertheless, existing methods operate on discrete sequences, lacking both continuous-time support and explicit conceptual separation between smooth trends and sparse events. These limitations motivate a unified framework that combines the continuous-time advantages of implicit neural

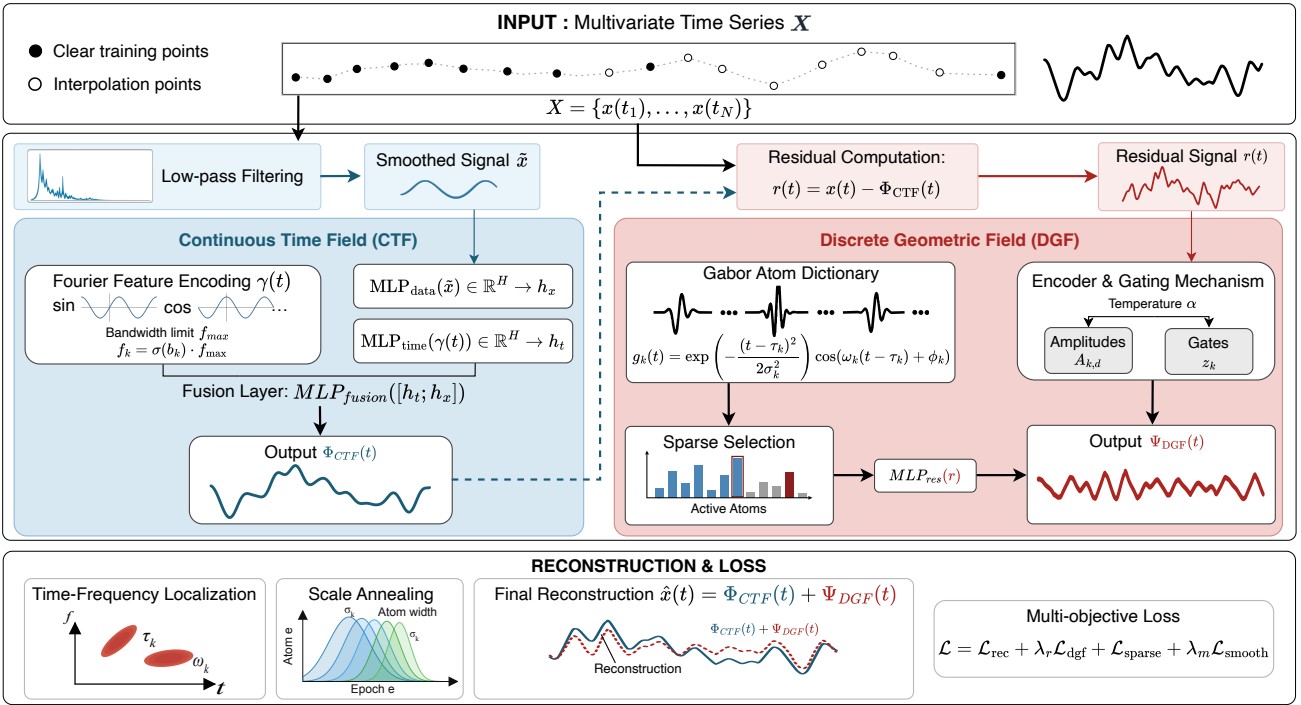

*Figure 1.* **DualTimesField architecture.** The CTF (blue) models smooth trends with bandwidth-limited encoding, while the DGF (red) captures sparse events using gated Gabor atoms.

fields with structured trend-event decomposition.

## 3. Methodology

Consider a multivariate time series $\mathbf{X} = \{\mathbf{x}(t_i)\}_{i=1}^{N}$ consisting of $N$ observations, where each observation $\mathbf{x}(t_i) \in \mathbb{R}^D$ is a $D$-dimensional vector recorded at timestamp $t_i$. We normalize all timestamps to the unit interval $[0, 1]$. DualTimesField learns a continuous generative function that decomposes the signal into two complementary components:

$$\hat{\mathbf{x}}(t) = \Phi_{\text{CTF}}(t; \theta_c) + \Psi_{\text{DGF}}(t; \theta_d) \tag{1}$$

where $\hat{\mathbf{x}}(t) \in \mathbb{R}^D$ denotes the reconstructed signal at time $t$, $\Phi_{\text{CTF}}$ is the Continuous Time Field parameterized by $\theta_c$, and $\Psi_{\text{DGF}}$ is the Discrete Geometric Field parameterized by $\theta_d$. The CTF captures smooth low-frequency trends that evolve gradually over time, while the DGF models sparse high-frequency transient events such as spikes, bursts, and abrupt transitions. This explicit decomposition addresses the spectral bias problem in standard implicit neural representations (Rahaman et al., 2019; Tancik et al., 2020), where multi-layer perceptrons preferentially fit low-frequency components while underfitting high-frequency structures. Figure 1 illustrates the overall architecture. The CTF uses bandwidth-limited temporal encoding to represent smooth trends, while the DGF employs learnable Gabor atoms with gated sparsity to capture localized events. The two fields are trained jointly to achieve decomposed continuous-time representations.

### 3.1. Bandwidth-Limited Trend Extraction

The CTF represents the smooth baseline of the signal through bandwidth-limited Fourier encoding, spectral filtering, and smoothness regularization.

We encode temporal coordinates using learnable bandwidth-limited Fourier features. For a timestamp $t \in [0, 1]$, the encoding is:

$$\gamma(t) = \big[ \cos(2\pi f_1 t), \sin(2\pi f_1 t), \dots,$$
$$\cos(2\pi f_K t), \sin(2\pi f_K t) \big]^\top \in \mathbb{R}^{2K} \tag{2}$$

where $K$ is the number of frequency components and $\{f_k\}_{k=1}^{K}$ are learnable frequencies. Unlike standard positional encodings (Tancik et al., 2020) that use fixed frequencies spanning a wide spectral range, we constrain each frequency below an explicit cutoff:

$$f_k = \sigma(b_k) \cdot f_{\max}, \quad k = 1, \dots, K \tag{3}$$

where $b_k \in \mathbb{R}$ is a learnable parameter, $\sigma(\cdot)$ is the sigmoid function ensuring $f_k \in (0, f_{\max})$, and $f_{\max}$ is the maximum allowable frequency. This prevents the CTF from representing high-frequency content that should be captured by the DGF.

Before encoding, we apply frequency-domain low-pass filtering to extract the smooth component:

$$\tilde{\mathbf{x}} = \mathcal{F}^{-1}\left[ H(f) \cdot \mathcal{F}[\mathbf{x}] \right] \tag{4}$$

where $\mathcal{F}[\cdot]$ denotes the discrete Fourier transform and $H(f)$ is a low-pass filter with smooth rolloff to avoid Gibbs artifacts (Oppenheim et al., 1999). The filter design is detailed in Appendix A.1.

The CTF processes temporal and data information through parallel encoding paths:

$$\mathbf{h}_t = \text{MLP}_{\text{time}}(\gamma(t)) \in \mathbb{R}^H \qquad (5)$$

$$\mathbf{h}_x = \text{MLP}_{\text{data}}(\tilde{\mathbf{x}}) \in \mathbb{R}^H \qquad (6)$$

where $H$ is the hidden dimension and each MLP consists of two linear layers with GELU activation (Hendrycks & Gimpel, 2016). The two representations are concatenated and fused:

$$\Phi_{\text{CTF}}(t) = \text{MLP}_{\text{fusion}}([\mathbf{h}_t; \mathbf{h}_x]) \in \mathbb{R}^D \qquad (7)$$

To encourage temporal smoothness, we impose a first-order regularization:

$$\mathcal{L}_{\text{smooth}} = \frac{1}{N-1} \sum_{i=1}^{N-1} \|\Phi_{\text{CTF}}(t_{i+1}) - \Phi_{\text{CTF}}(t_i)\|_2^2 \qquad (8)$$

### 3.2. Sparse Event Modeling with Gabor Atoms

The DGF models the residual signal $\mathbf{r}(t) = \mathbf{x}(t) - \Phi_{\text{CTF}}(t)$ using learnable Gabor atoms. Drawing on time-frequency analysis (Gabor, 1946; Daubechies, 1990) and sparse coding (Mairal et al., 2009), we represent the residual as a sparse superposition of basis functions that achieve optimal joint localization in time and frequency.

Each Gabor atom $g_k(t)$ is a modulated Gaussian wavelet:

$$g_k(t) = \exp\left(-\frac{(t-\tau_k)^2}{2\sigma_k^2}\right) \cos(\omega_k(t-\tau_k) + \phi_k) \qquad (9)$$

where $\tau_k \in [0,1]$ is the temporal center specifying when the event occurs, $\sigma_k > 0$ is the scale controlling event duration, $\omega_k > 0$ is the angular frequency, and $\phi_k$ is the phase. The Gaussian envelope localizes the atom in time around $\tau_k$, while the cosine carrier localizes it in frequency around $\omega_k$. This functional form achieves the minimum time-frequency uncertainty product according to the Heisenberg-Gabor principle (Gabor, 1946).

We maintain $M$ atoms with learnable parameters $\{\tau_k, \sigma_k, \omega_k, \phi_k\}_{k=1}^M$. To ensure valid ranges, we apply reparameterizations $\sigma_k = \text{softplus}(\tilde{\sigma}_k) + \epsilon$ and $\omega_k = \text{softplus}(\tilde{\omega}_k)$, where $\epsilon$ prevents degenerate zero-width atoms (see Appendix A.3).

Rather than using fixed amplitudes that would require separate optimization for each time series, we predict sample-specific coefficients conditioned on observed context. This design enables the model to adapt atom contributions based

on the input signal characteristics. We compute a context representation by averaging over observed values:

$$\bar{\mathbf{h}} = \frac{1}{N} \sum_{i=1}^N \mathbf{x}(t_i) \in \mathbb{R}^D \qquad (10)$$

This context summary is mapped to amplitudes and gates:

$$\mathbf{A} = \text{reshape}\left(\mathbf{W}_a \bar{\mathbf{h}} + \mathbf{b}_a\right) \in \mathbb{R}^{M \times D} \qquad (11)$$

$$\mathbf{z} = \sigma\left(\alpha \cdot \left(\mathbf{W}_z \bar{\mathbf{h}} + \mathbf{b}_z\right)\right) \in \mathbb{R}^M \qquad (12)$$

where $\mathbf{A}$ specifies the contribution of each atom to each variable, $\mathbf{z}$ controls atom activation, and $\alpha$ is a temperature parameter sharpening the gates toward binary selection. The gated amplitudes $\tilde{A}_{k,d} = A_{k,d} \cdot z_k$ allow selective atom activation.

Using these gated amplitudes, the DGF output is constructed as:

$$\Psi_{\text{DGF}}(t) = \sum_{k=1}^M \tilde{A}_{k,:} \cdot g_k(t) \qquad (13)$$

where amplitudes $\tilde{A}_{k,:}$ are derived from the context representation $\bar{\mathbf{h}}$, enabling inference without requiring ground truth at query time $t$.

To encourage sparse atom usage, we penalize the mean gate activation:

$$\mathcal{L}_{\text{sparse}} = \lambda_s \left(\bar{z} + \beta \cdot \text{ReLU}(\bar{z} - \rho)\right) \qquad (14)$$

where $\bar{z} = \frac{1}{M} \sum_{k=1}^M z_k$, $\rho$ is the target sparsity level, and $\beta$ penalizes exceeding this target.

### 3.3. Coarse-to-Fine Training with Scale Annealing

Training localized basis functions is challenging because narrow atoms create sharp, non-convex loss landscapes. We address this through scale annealing that progressively sharpens atom localization:

$$\tilde{\sigma}_k(e) = \sigma_k + \eta(e) \cdot \sigma_{\text{base}} \qquad (15)$$

where $e$ is the epoch, $\sigma_k$ is the learned scale, $\sigma_{\text{base}} = 0.1$, and $\eta(e) \in [0,1]$ is an annealing coefficient following a cosine schedule from 1 to 0. During warmup (first 30% of training), $\eta(e) = 1$ keeps atoms broad, creating a smooth loss landscape for coarse localization. After warmup, $\eta(e)$ decreases and atoms progressively sharpen to capture precise temporal boundaries. The complete schedule is in Appendix A.4. We initialize atom parameters from spectral analysis of the initial residual $\mathbf{r}^{(0)} = \mathbf{x} - \Phi_{\text{CTF}}^{(0)}$. Frequencies $\omega_k$ are set to dominant spectral peaks, and centers $\tau_k$ are placed at energy maxima with local suppression to ensure diversity (see Appendix A.5). This initialization substantially

accelerates convergence. The training objective combines reconstruction fidelity with regularization:

$$\mathcal{L} = \mathcal{L}_{\text{rec}} + \lambda_r \mathcal{L}_{\text{dgf}} + \mathcal{L}_{\text{sparse}} + \lambda_m \mathcal{L}_{\text{smooth}} \quad (16)$$

where $\mathcal{L}_{\text{rec}} = \text{MSE}(\hat{\mathbf{x}}, \mathbf{x})$ measures overall fidelity, and $\mathcal{L}_{\text{dgf}} = \text{MSE}(\Psi_{\text{DGF}}, \mathbf{r})$ supervises the DGF to fit the CTF residual, preventing the two fields from competing for the same signal components. Complete hyperparameters and optimization details are provided in Appendix A.7.

### 3.4. Continuous-Time Inference and Interpretability

Once trained, DualTimesField supports querying at arbitrary timestamps $t^* \in [0, 1]$:

$$\hat{\mathbf{x}}(t^*) = \Phi_{\text{CTF}}(t^*) + \Psi_{\text{DGF}}(t^*) \quad (17)$$

Inference uses the stored context representation $\bar{\mathbf{h}}$ from training observations to predict amplitudes, requiring only the time coordinate $t^*$ without ground truth values. This continuous-time formulation enables querying at finer temporal resolutions than training data, resampling irregular observations to uniform grids, and generating trajectories at arbitrary timestamps. The explicit decomposition supports interpretable analysis. The CTF output $\Phi_{\text{CTF}}(t)$ reveals smooth trends and baseline dynamics. The DGF output $\Psi_{\text{DGF}}(t)$ isolates transient events such as spikes and bursts. Each active atom (where $z_k > 0.5$) provides explicit event information: temporal location $\tau_k$, duration $\sigma_k$, and oscillation frequency $\omega_k$. This interpretability distinguishes DualTimesField from black-box approaches and enables domain experts to understand learned patterns.

## 4. Experiments

We evaluate DualTimesField on three aspects: (1) whether explicit trend-event decomposition improves reconstruction fidelity, (2) whether the continuous-time formulation supports interpolation on irregularly sampled data, and (3) whether the learned decomposition provides interpretable components. These experiments jointly validate our dual-field design and its key components.

### 4.1. Experimental Details

**Datasets.** We evaluate DualTimesField on nine long-horizon benchmarks spanning power systems, energy consumption, finance, traffic, weather, and public health, including ETT (Zhou et al., 2021) with ETTh1, ETTh2, ETTm1, and ETTm2, as well as Electricity (Trindade, 2015), Exchange (Lai et al., 2018), Traffic (Cuturi, 2011), Weather (Max Planck Institute for Biogeochemistry), and ILI (Centers for Disease Control and Prevention, 2025).

For irregular sampling, we evaluate on six benchmarks spanning diverse domains and irregularity mechanisms.

PhysioNet (Silva et al., 2012) contains multivariate ICU records with missing and asynchronous observations, and USHCN (Menne et al., 2016) provides long-term climate measurements with irregular observation masks, both following the protocol of (De Brouwer et al., 2019). Human Activity records 3D body-sensor positions with asynchronous timestamps under the setup of (Shukla & Marlin, 2021). We additionally include EPA-Air, ClusterTrace, and FN-SPID from the Time-IMM benchmark (Chang et al., 2025), which together capture constraint-based and artifact-based irregularity patterns from real-world systems.

**Metrics.** The fidelity of reconstruction and interpolation is assessed using mean squared error (MSE) and mean absolute error (MAE) calculated between the model outputs and ground-truth observations across all variables and timestamps.

**Baselines.** For reconstruction, we compare DualTimesField against three categories of methods: implicit neural representations, including SIREN (Sitzmann et al., 2020), WIRE (Saragadam et al., 2023), and Fourier feature networks (Tancik et al., 2020); deep time series models trained for self-reconstruction, including N-BEATS (Oreshkin et al., 2020), PatchTST (Nie et al., 2023), iTransformer (Liu et al., 2024) and TimesNet (Wu et al., 2023); and classical dimensionality reduction via PCA (Jolliffe & Cadima, 2016). For interpolation on irregularly sampled data, we compare against established baselines for sparse observations, including RNN (Rumelhart et al., 1985), RNN-VAE (Chung et al., 2014), ODE-RNN (Rubanova et al., 2019), GRU-D (Che et al., 2018), Latent ODE (Chen et al., 2018; Rubanova et al., 2019), SAITS (Du et al., 2023), CRU (Schirmer et al., 2022), LS4 (Zhou et al., 2023).

### 4.2. Reconstruction Performance

DualTimesField explicitly separates a bandwidth-limited trend field from a sparse event field, targeting high-fidelity continuous-time representation. To evaluate this, we measure reconstruction error on nine long-horizon benchmarks.

Table 1 demonstrates that DualTimesField achieves the best MSE across all nine benchmarks and the best MAE on eight of them, reducing MSE by 51.2% and MAE by 31.7% on average compared with TimesNet, the strongest deep-learning baseline overall. The improvements are particularly substantial on datasets with rich temporal dynamics: ETTm2, Weather, and ETTh2 achieve 96.9%, 75.0%, and 59.5% MSE reduction respectively, with corresponding MAE reductions of 71.3%, 45.3%, and 31.6%. On high-dimensional Electricity (321 variables), DualTimesField achieves an MSE of 0.1322 compared with 0.9791 for Fourier features. Even the smallest improvement of 11.4% on Traffic represents meaningful error reduction, while the largest gains approach order-of-magnitude improvements. The sole ex-

*Table 1.* Time-series reconstruction results (mean ± std over 5 random seeds). **Best** and second-best results are highlighted.

| Dataset | Metric | SIREN | WIRE | Fourier | PCA | N-BEATS | PatchTST | iTransformer | TimesNet | **Ours** |
|---|---|---|---|---|---|---|---|---|---|---|
| ETTh1 | MSE ↓ | $1.0011_{\pm0.0014}$ | $1.0067_{\pm0.0016}$ | $1.0040_{\pm0.0004}$ | $0.5165_{\pm0.0032}$ | $0.8264_{\pm0.0136}$ | $0.6606_{\pm0.0046}$ | $0.2439_{\pm0.0022}$ | $0.0155_{\pm0.0004}$ | $\mathbf{0.0084}_{\pm0.0008}$ |
| | MAE ↓ | $0.8129_{\pm0.0020}$ | $0.8154_{\pm0.0013}$ | $0.8146_{\pm0.0002}$ | $0.5499_{\pm0.0022}$ | $0.7229_{\pm0.0093}$ | $0.6342_{\pm0.0049}$ | $0.3680_{\pm0.0014}$ | $0.0863_{\pm0.0028}$ | $\mathbf{0.0655}_{\pm0.0041}$ |
| ETTh2 | MSE ↓ | $0.8412_{\pm0.0010}$ | $0.8487_{\pm0.0032}$ | $0.8395_{\pm0.0002}$ | $0.5071_{\pm0.0102}$ | $0.7818_{\pm0.0107}$ | $0.5694_{\pm0.0028}$ | $0.2424_{\pm0.0030}$ | $0.0975_{\pm0.0101}$ | $\mathbf{0.0395}_{\pm0.0046}$ |
| | MAE ↓ | $0.6921_{\pm0.0006}$ | $0.6960_{\pm0.0006}$ | $0.6951_{\pm0.0001}$ | $0.5363_{\pm0.0042}$ | $0.6707_{\pm0.0048}$ | $0.5684_{\pm0.0025}$ | $0.3689_{\pm0.0022}$ | $0.1987_{\pm0.0154}$ | $\mathbf{0.1360}_{\pm0.0099}$ |
| ETTm1 | MSE ↓ | $0.9958_{\pm0.0015}$ | $1.0005_{\pm0.0005}$ | $0.9989_{\pm0.0005}$ | $0.3115_{\pm0.0012}$ | $0.6975_{\pm0.0033}$ | $0.3060_{\pm0.0006}$ | $0.0624_{\pm0.0003}$ | $0.0022_{\pm0.0002}$ | $\mathbf{0.0013}_{\pm0.0001}$ |
| | MAE ↓ | $0.8140_{\pm0.0006}$ | $0.8148_{\pm0.0005}$ | $0.8145_{\pm0.0001}$ | $0.4185_{\pm0.0012}$ | $0.6535_{\pm0.0020}$ | $0.4209_{\pm0.0022}$ | $0.1677_{\pm0.0005}$ | $0.0297_{\pm0.0024}$ | $\mathbf{0.0222}_{\pm0.0021}$ |
| ETTm2 | MSE ↓ | $0.9435_{\pm0.0008}$ | $0.9434_{\pm0.0002}$ | $0.9427_{\pm0.0001}$ | $0.3045_{\pm0.0015}$ | $0.7825_{\pm0.0053}$ | $0.3378_{\pm0.0012}$ | $0.0824_{\pm0.0009}$ | $0.2424_{\pm0.0350}$ | $\mathbf{0.0075}_{\pm0.0006}$ |
| | MAE ↓ | $0.7322_{\pm0.0009}$ | $0.7322_{\pm0.0001}$ | $0.7324_{\pm0.0001}$ | $0.4132_{\pm0.0009}$ | $0.6630_{\pm0.0024}$ | $0.4399_{\pm0.0031}$ | $0.2016_{\pm0.0011}$ | $0.1851_{\pm0.0087}$ | $\mathbf{0.0531}_{\pm0.0036}$ |
| Electricity | MSE ↓ | $0.9799_{\pm0.0002}$ | $0.9791_{\pm0.0002}$ | $0.9791_{\pm0.0001}$ | $0.5132_{\pm0.0023}$ | $0.7562_{\pm0.0103}$ | $0.2542_{\pm0.0002}$ | $0.1958_{\pm0.0007}$ | $0.1836_{\pm0.0033}$ | $\mathbf{0.1322}_{\pm0.0010}$ |
| | MAE ↓ | $0.8329_{\pm0.0006}$ | $0.8315_{\pm0.0004}$ | $0.8321_{\pm0.0001}$ | $0.5595_{\pm0.0014}$ | $0.6939_{\pm0.0063}$ | $0.3799_{\pm0.0012}$ | $0.3350_{\pm0.0012}$ | $0.3127_{\pm0.0031}$ | $\mathbf{0.2699}_{\pm0.0012}$ |
| Exchange | MSE ↓ | $0.8807_{\pm0.0011}$ | $0.8861_{\pm0.0114}$ | $0.8401_{\pm0.0005}$ | $0.2387_{\pm0.0030}$ | $0.3899_{\pm0.0125}$ | $0.1468_{\pm0.0008}$ | $0.0556_{\pm0.0006}$ | $0.1052_{\pm0.0401}$ | $\mathbf{0.0460}_{\pm0.0031}$ |
| | MAE ↓ | $0.8025_{\pm0.0004}$ | $0.8037_{\pm0.0025}$ | $0.7795_{\pm0.0003}$ | $0.3642_{\pm0.0039}$ | $0.4867_{\pm0.0107}$ | $0.2669_{\pm0.0017}$ | $\mathbf{0.1325}_{\pm0.0008}$ | $0.2377_{\pm0.0406}$ | $0.1415_{\pm0.0036}$ |
| Weather | MSE ↓ | $1.0101_{\pm0.0001}$ | $1.0105_{\pm0.0005}$ | $1.0095_{\pm0.0002}$ | $0.4270_{\pm0.0033}$ | $0.6253_{\pm0.0029}$ | $0.1035_{\pm0.0059}$ | $0.0870_{\pm0.0005}$ | $0.0440_{\pm0.0030}$ | $\mathbf{0.0110}_{\pm0.0026}$ |
| | MAE ↓ | $0.7459_{\pm0.0010}$ | $0.7463_{\pm0.0008}$ | $0.7443_{\pm0.0001}$ | $0.4138_{\pm0.0015}$ | $0.5143_{\pm0.0019}$ | $0.1490_{\pm0.0076}$ | $0.1193_{\pm0.0004}$ | $0.1358_{\pm0.0061}$ | $\mathbf{0.0743}_{\pm0.0100}$ |
| ILI | MSE ↓ | $1.0260_{\pm0.0033}$ | $1.0125_{\pm0.0310}$ | $1.0297_{\pm0.0009}$ | $0.3590_{\pm0.0055}$ | $0.4827_{\pm0.0190}$ | $0.4875_{\pm0.0036}$ | $0.1982_{\pm0.0135}$ | $0.0962_{\pm0.0191}$ | $\mathbf{0.0510}_{\pm0.0033}$ |
| | MAE ↓ | $0.8004_{\pm0.0019}$ | $0.7910_{\pm0.0073}$ | $0.7989_{\pm0.0004}$ | $0.4382_{\pm0.0055}$ | $0.5089_{\pm0.0121}$ | $0.5176_{\pm0.0046}$ | $0.3025_{\pm0.0124}$ | $0.2227_{\pm0.0220}$ | $\mathbf{0.1562}_{\pm0.0087}$ |
| Traffic | MSE ↓ | $1.0650_{\pm0.0047}$ | $1.0214_{\pm0.0001}$ | $1.0222_{\pm0.0001}$ | $0.6661_{\pm0.0060}$ | $0.8032_{\pm0.0145}$ | $0.6046_{\pm0.0021}$ | $0.5010_{\pm0.0041}$ | $0.2382_{\pm0.0017}$ | $\mathbf{0.2111}_{\pm0.0015}$ |
| | MAE ↓ | $0.7509_{\pm0.0016}$ | $0.7384_{\pm0.0001}$ | $0.7388_{\pm0.0001}$ | $0.5586_{\pm0.0038}$ | $0.6101_{\pm0.0071}$ | $0.5082_{\pm0.0012}$ | $0.5008_{\pm0.0032}$ | $0.2654_{\pm0.0011}$ | $\mathbf{0.2559}_{\pm0.0011}$ |

ception is the Exchange MAE, where iTransformer reaches 0.1325 against our 0.1415. Since our method retains the best MSE on this dataset, the gap reflects a looser fit on small fluctuations rather than missed structure.

The consistent superiority validates that explicit trend-event separation effectively addresses the spectral bias and frequency entanglement inherent in single-field INRs. Standard INR methods including SIREN, WIRE, and Fourier features exhibit uniformly high errors around 0.840–1.065 MSE, while deep learning baselines show moderate improvements, with N-BEATS achieving 0.390–0.826 MSE, PatchTST 0.104–0.661 MSE, iTransformer 0.056–0.501 MSE, and TimesNet further reducing errors to 0.002–0.242 MSE. DualTimesField consistently outperforms all three categories by structurally preventing trend-event interference rather than relying on implicit learning to disentangle frequency components, confirming that structured frequency separation provides a more natural inductive bias for complex time series than holistic implicit representations.

### 4.3. Evaluation of Temporal Interpolation

Beyond reconstruction quality, we evaluate DualTimesField's interpolation performance on irregularly sampled data with sparse, unevenly distributed observations. This tests whether the learned trend-event decomposition generalizes to unobserved temporal locations. DualTimesField achieves the lowest MSE on five of the six datasets, with margins exceeding 90% over the second-best method on EPA-Air and Human Activity.

*Table 2.* Interpolation performance (MSE $\times 10^{-3}$).

| Method | EPA-Air | FNSPID | ClusterTrace |
|---|---|---|---|
| RNN | 8.62 | 2.21 | 4.54 |
| RNN-VAE | 17.58 | 32.98 | 12.25 |
| ODE-RNN | 12.33 | 1.15 | 2.69 |
| GRU-D | 8.97 | 2.44 | 3.71 |
| Latent ODE | 6.67 | 1.67 | 5.46 |
| SAITS | 17.36 | 11.89 | 4.27 |
| CRU | 10.09 | 10.47 | 9.78 |
| LS4 | 2.17 | 5.92 | 3.78 |
| **Ours** | **0.15** | **0.58** | **1.82** |

| Method | Human Activity | PhysioNet | USHCN |
|---|---|---|---|
| RNN | 5.42 | 2.92 | 4.32 |
| RNN-VAE | 17.88 | 5.93 | 7.56 |
| ODE-RNN | 11.67 | 2.23 | 2.47 |
| GRU-D | 5.44 | 3.33 | 3.40 |
| Latent ODE | 7.26 | 8.34 | 6.86 |
| SAITS | 11.24 | 10.70 | 3.61 |
| CRU | 15.51 | 1.82 | 0.16 |
| LS4 | 4.24 | 0.63 | **0.06** |
| **Ours** | **0.19** | **0.30** | 0.17 |

Table 2 shows interpolation MSE scores in units of $10^{-3}$

on six irregularly sampled benchmarks, where models are trained on sparse observations and evaluated on withheld timestamps. DualTimesField attains the lowest MSE on five of the six datasets. The most pronounced gain occurs on PhysioNet, where it achieves $0.30 \times 10^{-3}$, outperforming the strongest baseline LS4 by 52.4% and substantially exceeding neural ODE methods such as ODE-RNN and Latent ODE. On USHCN, however, LS4 and CRU obtain the two leading results at $0.06 \times 10^{-3}$ and $0.16 \times 10^{-3}$ respectively, compared with $0.17 \times 10^{-3}$ for DualTimesField. Although DualTimesField still outperforms most baselines on USHCN, this performance gap reveals an important characteristic of our approach. PhysioNet contains complex physiological signals with abrupt events such as heart rate spikes and irregular vital sign fluctuations, where explicit trend-event decomposition provides clear advantages by separately modeling the smooth baseline and transient deviations. In contrast, USHCN climate data exhibits slowly-varying seasonal patterns with fewer localized transient events, where the DGF's sparse Gabor atoms have limited contributions and continuous state-space methods such as LS4 and CRU can more efficiently capture the smooth temporal evolution. These results suggest that DualTimesField is particularly effective for time series with rich event structure superimposed on smooth trends, while purely trend-dominated signals may benefit more from methods specifically designed for smooth dynamics. This characterization aligns with our design motivation of explicitly separating trends from events.

## 4.4. Synthetic Sanity Check

We construct a synthetic signal with known ground truth decomposition. A test signal is generated, which comprises smooth trends formed by a mixture of low-frequency sinusoids below 5 Hz and sparse transient events consisting of localized Gabor-like bursts above 20 Hz, combined with additive Gaussian noise. This controlled setup tests whether CTF and DGF isolate their respective frequency ranges and whether single-field representations exhibit the spectral bias motivating our design.

Figure 2 shows that the CTF recovers the smooth trends with Pearson correlation 0.666, demonstrating effective low-frequency capture. More critically, the DGF successfully detects all five events with mean temporal error $< 0.02$, confirming precise event localization despite its amplitude correlation of 0.117, which reflects the design's prioritization of temporal accuracy in sparse event modeling. Spectral analysis confirms effective separation. The CTF concentrates 90% of its energy below the 5 Hz cutoff, while the DGF retains 51% above 20 Hz. In contrast, a single-field implicit neural representation achieves substantially higher reconstruction error and exhibits energy leakage across the frequency boundary, failing to achieve structured decom-

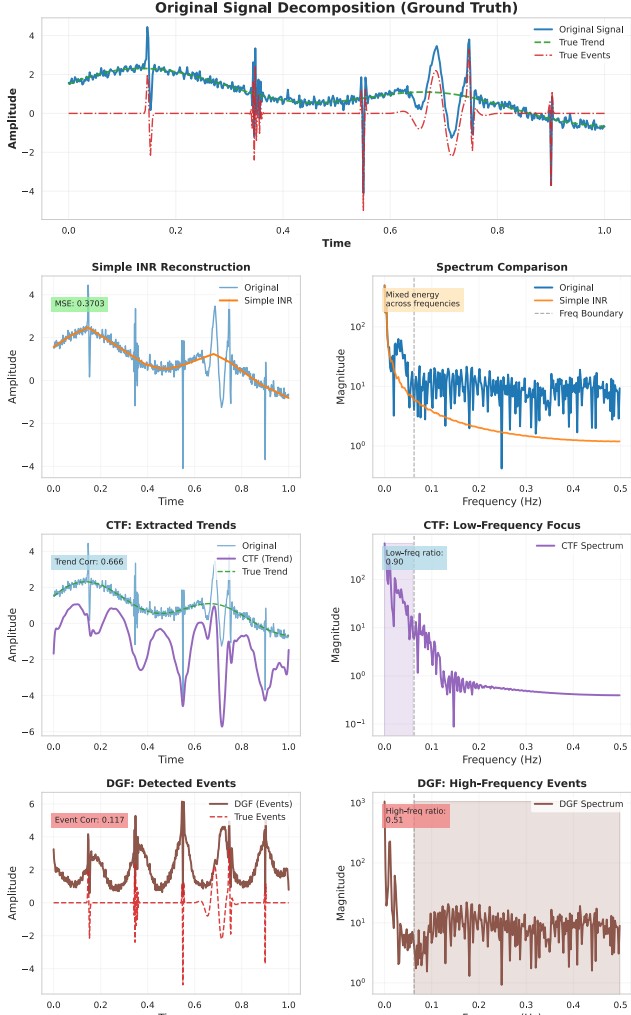

*Figure 2.* Synthetic sanity check. Left: time domain; right: spectra. Top to bottom: ground truth, single-field INR, CTF trend, and DGF events. The dashed line marks the frequency boundary.

position. These results validate that bandwidth-limited parameterization effectively enforces frequency separation, preventing the trend-event interference inherent in single-field representations.

## 4.5. Ablation Analysis

To validate the rationality of the proposed dual-field decomposition, we compare the three model variants. Among them, **CTF-only** uses only the Continuous Time Field to model smooth trends, **DGF-only** relies solely on the Discrete Geometric Field to capture sparse events, and the **Full** model combines both CTF and DGF.

Table 3 shows that, combining both the CTF and DGF fields (Full model) yields the best overall reconstruction accuracy. On average, the Full model reduces MSE by 53.5% and MAE by 39.9% relative to CTF-only, and MSE by 21.1%

and MAE by 15.5% relative to DGF-only, with the largest gain on ETTm1 where MSE drops from 0.042 to 0.001. The Full model attains the best MAE on all nine datasets and the best MSE on seven, with the remaining two MSE differences of 0.001. This pattern indicates that modeling trends alone cannot recover high-frequency events, while modeling events alone is less effective without a continuous baseline to capture global structure.

*Table 3.* Ablation analysis on dual-field decomposition (reconstruction error).

| Dataset | MSE ↓ | | | MAE ↓ | | |
|---|---|---|---|---|---|---|
| | CTF | DGF | Full | CTF | DGF | Full |
| ETTh1 | 0.112 | 0.011 | **0.008** | 0.231 | 0.077 | **0.066** |
| ETTh2 | 0.130 | **0.039** | 0.040 | 0.256 | 0.139 | **0.136** |
| ETTm1 | 0.042 | 0.002 | **0.001** | 0.127 | 0.034 | **0.022** |
| ETTm2 | 0.057 | 0.012 | **0.008** | 0.159 | 0.070 | **0.053** |
| Electricity | 0.198 | 0.158 | **0.132** | 0.316 | 0.289 | **0.270** |
| Exchange | 0.099 | 0.113 | **0.046** | 0.215 | 0.247 | **0.142** |
| Weather | 0.097 | 0.035 | **0.011** | 0.164 | 0.124 | **0.074** |
| ILI | 0.103 | **0.050** | 0.051 | 0.220 | 0.156 | **0.156** |
| Traffic | 0.255 | 0.216 | **0.211** | 0.272 | 0.256 | **0.256** |
| Average | 0.121 | 0.071 | **0.056** | 0.218 | 0.155 | **0.131** |

Beyond reconstruction accuracy, Table 4 quantifies how the event field is utilized through the number of active Gabor atoms, which varies with dataset complexity. In the full model, CTF absorbs smooth components and can reduce DGF usage on smoother data, for example Weather decreases from 3.0 to 2.5 and Traffic decreases from 8.0 to 6.0. In contrast, on more event-rich datasets, CTF captures global structure and exposes localized residual events, leading DGF to activate more atoms, for example ETTh1 increases from 2.0 to 5.5 and ILI increases from 5.5 to 9.0. This data-dependent sparsity confirms adaptive capacity allocation between the two fields.

*Table 4.* Average number of active atoms in the DGF-only and Full model.

| Dataset | Active Atoms (DGF / Full) |
|---|---|
| ETTh1 | 2.0 / 5.5 |
| ETTh2 | 5.5 / 7.5 |
| ETTm1 | 1.5 / 4.0 |
| ETTm2 | 4.5 / 4.0 |
| Electricity | 6.5 / 4.5 |
| Exchange | 8.0 / 9.0 |
| Weather | 3.0 / 2.5 |
| ILI | 5.5 / 9.0 |
| Traffic | 8.0 / 6.0 |

The ablations reveal clear field specialization. CTF captures global smooth structure, while DGF allocates sparse atoms

to explain transient residuals in a data-dependent manner. This complementarity explains the consistent error reduction of the full model and supports explicit trend–event decomposition as an effective inductive bias.

### 4.6. Event Interpretability

To demonstrate that explicit trend-event decomposition yields interpretable components consistent with heartbeat-specific transients, we evaluate DualTimesField on the ECG5000 dataset (Dau et al., 2019). This dataset contains 5000 single-heartbeat electrocardiogram segments of length 140 time steps across five heartbeat types, including Normal (N), R-on-T Premature Ventricular Contraction (r), Supraventricular (S), Premature Ventricular Contraction (V), and Unclassifiable (Q).

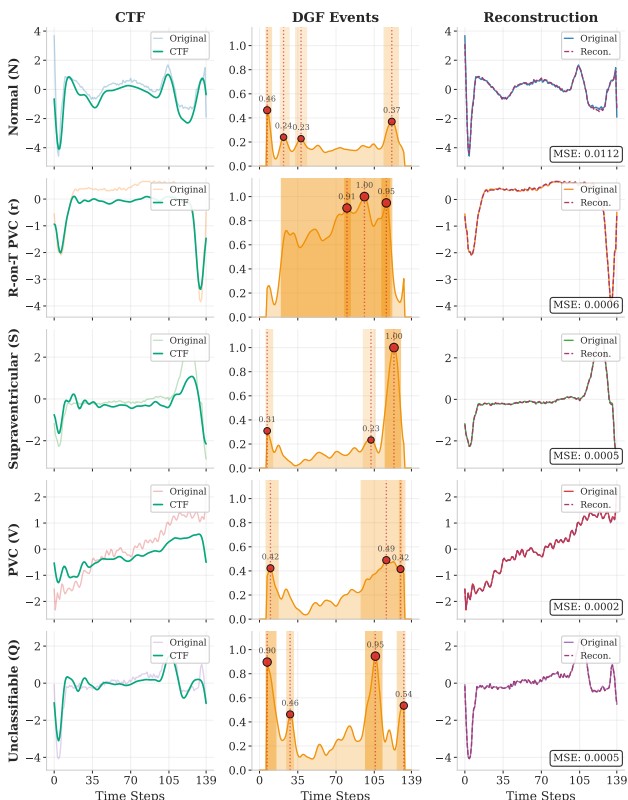

*Figure 3.* Interpretable trend-event decomposition on ECG5000. Left: CTF baseline overlaid on original waveform. Middle: DGF event saliency with activated Gabor atoms marked. Right: final reconstruction.

Figure 3 shows representative decompositions for one example per category, with all reconstructions achieving MSE below $1.2 \times 10^{-2}$. The decomposition provides qualitative evidence that DualTimesField separates smooth morphology from transient deviations in a class-consistent and parsimonious manner. For Normal beats, event saliency remains minimal and the CTF baseline closely tracks the waveform, indicating that the model allocates representational capacity

primarily to the trend field when transient structure is absent. This validates that the DGF does not spuriously activate on smooth signals. In contrast, arrhythmic beats trigger pronounced and spatially localized event responses that align with clinical characteristics. R-on-T Premature Ventricular Contraction exhibits concentrated late-phase activation corresponding to the abnormal ventricular depolarization. Supraventricular arrhythmia demonstrates dual activation peaks reflecting the characteristic double-bump morphology. Premature Ventricular Contraction activates multiple distributed atoms across the entire waveform, capturing its complex multi-component transient structure. Critically, these event patterns emerge without supervision on heartbeat categories, demonstrating that explicit trend-event decomposition discovers clinically meaningful structure purely from reconstruction objectives.

## 5. Conclusion

We propose DUALTIMESFIELD, a continuous-time representation framework that decomposes time series into smooth trends and sparse events via dual implicit neural fields. This design addresses the spectral bias and frequency entanglement inherent in single-field representations, achieving 51.2% average MSE reduction across nine benchmarks. Ablation studies confirm that neither field alone matches the full model, while analysis on ECG5000 demonstrates clinically meaningful decomposition without supervision. These findings validate explicit trend-event decomposition as an effective inductive bias for time series with rich event structure. Future work could explore hierarchical multi-scale extensions, transferability to downstream tasks, and adaptive frequency selection. By bridging classical signal decomposition with modern implicit representations, DualTimesField provides a principled approach to interpretable and high-fidelity temporal modeling.

## Acknowledgements

This work was supported by the National Natural Science Foundation of China (Grant No. 62462059).

## Impact Statement

This paper presents work whose goal is to advance the field of time series representation learning. All datasets used in our experiments are publicly available benchmarks from domains including healthcare, energy, transportation, and climate. We believe DualTimesField could support beneficial applications in these domains, while acknowledging that clinical deployment would require rigorous validation and regulatory approval, and that responsible deployment more broadly requires appropriate human oversight and domain expertise.

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

## A. Notation

This section summarizes the symbols and notations used throughout the paper for ease of reference. Table 5 lists all variables and their definitions unless otherwise specified.

*Table 5.* Summary of symbols and notations used in DualTimesField.

| Symbol | Description | Symbol | Description |
|---|---|---|---|
| $t \in [0, 1]$ | Continuous time variable | $r(t)$ | Residual signal, $x(t) - \Phi_{\mathrm{CTF}}(t)$ |
| $t_i$ | Timestamp of the $i$-th observation | $M$ | Number of Gabor atoms in the DGF |
| $N$ | Number of observed time points | $g_k(t)$ | $k$-th Gabor atom |
| $D$ | Dimensionality of the time series | $\tau_k$ | Temporal center of the $k$-th atom |
| $x(t)$ | Ground-truth multivariate time series | $\sigma_k$ | Temporal scale (width) of the atom |
| $x(t_i)$ | Observed value at timestamp $t_i$ | $\omega_k$ | Angular frequency of the atom |
| $\hat{x}(t)$ | Model reconstruction at time $t$ | $\phi_k$ | Phase of the Gabor atom |
| $\Phi_{\mathrm{CTF}}(t)$ | Continuous Time Field (CTF) | $A \in \mathbb{R}^{M \times D}$ | Atom amplitude matrix |
| $\Psi_{\mathrm{DGF}}(t)$ | Discrete Geometric Field (DGF) | $z_k$ | Gating variable for atom $k$ |
| $\theta_c$ | Parameters of the CTF | $\tilde{A}_{k,d}$ | Gated amplitude for variable $d$ |
| $\theta_d$ | Parameters of the DGF | $\bar{z}$ | Mean activation level of atom gates |
| $\gamma(t)$ | Fourier-based temporal encoding | $\mathcal{L}_{\mathrm{rec}}$ | Reconstruction loss (MSE) |
| $K$ | Number of Fourier components | $\mathcal{L}_{\mathrm{dgf}}$ | Residual reconstruction loss |
| $f_k$ | Learnable frequency of $k$-th feature | $\mathcal{L}_{\mathrm{smooth}}$ | Temporal smoothness regularization |
| $f_{\max}$ | Max allowed frequency for CTF | $\mathcal{L}_{\mathrm{sparse}}$ | Sparsity regularization |
| $H(f)$ | Low-pass frequency-domain filter | $\lambda_r, \lambda_m, \lambda_s$ | Weights for DGF, smooth, and sparse |
| $\mathcal{F}[\cdot]$ | Discrete Fourier transform | $\eta(e)$ | Scale annealing coefficient at epoch $e$ |
| $\mathcal{F}^{-1}[\cdot]$ | Inverse discrete Fourier transform | $E$ | Total number of training epochs |
| $\tilde{x}$ | Low-pass filtered signal | $\sigma_{\mathrm{base}}$ | Base scale for annealing |

This appendix provides complete implementation details for DualTimesField, including filter design, parameter specifications, training schedules, and initialization procedures.

### A.1. Frequency-Domain Filter Design

The low-pass filter $H(f)$ in the CTF is designed with a smooth rolloff to avoid Gibbs phenomenon (ringing artifacts) that arise from ideal brick-wall filters. The filter is defined as:

$$H(f) = \begin{cases} 1 & \text{if } f < f_c \\ 1 - \frac{f - f_c}{f_c} & \text{if } f_c \le f < 2f_c \\ 0 & \text{if } f \ge 2f_c \end{cases} \tag{18}$$

where $f_c = N/16$ is the cutoff frequency index, corresponding to retaining the lowest 1/8 of frequency components. The passband $[0, f_c)$ has unit gain, preserving low-frequency content without attenuation. The transition band $[f_c, 2f_c)$ tapers linearly from 1 to 0, providing a gradual rolloff that avoids the sharp discontinuity of ideal filters. The stopband $[2f_c, \infty)$ has zero gain, completely removing high-frequency content. This design ensures that the filtered signal $\tilde{x}$ contains only smooth, slowly-varying components suitable for CTF processing.

### A.2. Network Architecture and Initialization

All networks use a hidden dimension $H = 64$ and GELU activation. The CTF time encoder transforms Fourier features through two linear layers:

$$\mathbf{h}_t = \mathrm{GELU}\left(\mathbf{W}_2^{(t)} \cdot \mathrm{GELU}\left(\mathbf{W}_1^{(t)}\gamma(t) + \mathbf{b}_1^{(t)}\right) + \mathbf{b}_2^{(t)}\right) \tag{19}$$

where $\mathbf{W}_1^{(t)} \in \mathbb{R}^{H \times 2K}$ projects from $2K$-dimensional Fourier features to hidden dimension, and $\mathbf{W}_2^{(t)} \in \mathbb{R}^{H \times H}$ is a square transformation. The data encoder has analogous structure with $\mathbf{W}_1^{(x)} \in \mathbb{R}^{H \times D}$ mapping from input dimension $D$. The fusion network concatenates the two hidden vectors and produces the output:

$$\Phi_{\mathrm{CTF}}(t) = \mathbf{W}_4 \cdot \mathrm{GELU}\left(\mathbf{W}_3[\mathbf{h}_t; \mathbf{h}_x] + \mathbf{b}_3\right) + \mathbf{b}_4 \tag{20}$$

where $\mathbf{W}_3 \in \mathbb{R}^{H \times 2H}$ and $\mathbf{W}_4 \in \mathbb{R}^{D \times H}$.

The DGF encoder processes the residual signal:

$$\mathbf{h} = \text{GELU}\left(\mathbf{W}_6 \cdot \text{GELU}\left(\mathbf{W}_5 \mathbf{r} + \mathbf{b}_5\right) + \mathbf{b}_6\right) \in \mathbb{R}^{N \times H} \tag{21}$$

with $\mathbf{W}_5 \in \mathbb{R}^{H \times D}$ and $\mathbf{W}_6 \in \mathbb{R}^{H \times H}$. The amplitude projection uses $\mathbf{W}_a \in \mathbb{R}^{(M \cdot D) \times H}$ reshaped to $M \times D$, and the gate projection uses $\mathbf{W}_z \in \mathbb{R}^{M \times H}$. The residual correction network uses $\mathbf{W}_7 \in \mathbb{R}^{H \times D}$ and $\mathbf{W}_8 \in \mathbb{R}^{D \times H}$.

All weight matrices are initialized using Kaiming uniform initialization, and bias vectors are initialized to zero. The learnable frequency parameters $\{b_k\}$ are initialized from $\mathcal{U}(-2, 2)$.

## A.3. Parameter Reparameterization

To ensure valid parameter ranges during optimization, we apply the following reparameterizations for Gabor atom parameters:

$$\sigma_k = \text{softplus}(\tilde{\sigma}_k) + \epsilon, \quad \epsilon = 0.02 \tag{22}$$

$$\omega_k = \text{softplus}(\tilde{\omega}_k) \tag{23}$$

where $\tilde{\sigma}_k, \tilde{\omega}_k \in \mathbb{R}$ are unconstrained learnable parameters and $\text{softplus}(x) = \log(1 + e^x)$ ensures positive outputs. The constant $\epsilon = 0.02$ prevents degenerate zero-width atoms that would cause numerical instability and infinite gradients. The temporal centers $\tau_k$ and phases $\phi_k$ are directly parameterized without reparameterization.

## A.4. Cosine Annealing Schedule

The annealing coefficient $\eta(e)$ follows a cosine schedule:

$$\eta(e) = \begin{cases} 1 & \text{if } e < e_{\text{warm}} \\ \frac{1}{2}\left(1 + \cos\left(\pi \cdot \frac{e - e_{\text{warm}}}{E - e_{\text{warm}}}\right)\right) & \text{if } e \geq e_{\text{warm}} \end{cases} \tag{24}$$

where $e$ is the current epoch, $e_{\text{warm}} = 0.3E$ is the warmup period (first 30% of training), and $E$ is the total number of epochs. During warmup, $\eta(e) = 1$ keeps atoms broad with effective scale $\tilde{\sigma}_k \approx \sigma_k + \sigma_{\text{base}}$ where $\sigma_{\text{base}} = 0.1$. This creates a smooth loss landscape that allows the model to first identify approximate event locations through gradient descent. After warmup, $\eta(e)$ decreases smoothly from 1 to 0 following a cosine curve, and atoms progressively sharpen to capture precise temporal boundaries. At the end of training ($e = E$), $\eta(E) = 0$ and atoms operate at their learned scales $\sigma_k$.

## A.5. Spectral-Guided Initialization

We initialize atom parameters from spectral analysis of an initial residual estimate. After a forward pass through the randomly initialized CTF, we compute $\mathbf{r}^{(0)} = \mathbf{x} - \Phi_{\text{CTF}}^{(0)}$ and analyze its frequency content.

Atom frequencies are initialized to dominant spectral peaks:

$$\omega_k^{(0)} = \frac{2\pi}{N} \cdot \text{argtop}_k\left(\frac{1}{BD} \sum_{b=1}^{B} \sum_{d=1}^{D} \left|\mathcal{F}[\mathbf{r}^{(0)}]_{b,:,d}\right|\right) \tag{25}$$

where $B$ is batch size, $\mathcal{F}[\mathbf{r}^{(0)}]_{b,:,d}$ is the Fourier transform of the $d$-th variable in the $b$-th sample, and $\text{argtop}_k(\cdot)$ returns the index of the $k$-th largest value.

Temporal centers are initialized at successive energy peaks with local suppression:

$$\tau_k^{(0)} = \frac{1}{N} \arg\max_i \left\{ \|\mathbf{r}^{(0)}(t_i)\|_2^2 \cdot \prod_{j=1}^{k-1} \mathbf{1}\left[\left|i - N\tau_j^{(0)}\right| > \frac{N}{2M}\right] \right\} \tag{26}$$

The indicator function enforces minimum separation of $N/(2M)$ samples between adjacent centers, ensuring spatial diversity.

When spectral initialization is not used, we apply uniform initialization:

$$\tau_k^{(0)} = 0.05 + 0.9 \cdot \frac{k-1}{M-1} \tag{27}$$

$$\omega_k^{(0)} = 10 + 90 \cdot \frac{k-1}{M-1} \tag{28}$$

### A.6. Multi-Objective Loss Formulation

The complete loss function is:

$$\mathcal{L} = \mathcal{L}_{\text{rec}} + \lambda_r \mathcal{L}_{\text{dgf}} + \mathcal{L}_{\text{sparse}} + \lambda_m \mathcal{L}_{\text{smooth}} \tag{29}$$

The reconstruction loss measures overall fidelity:

$$\mathcal{L}_{\text{rec}} = \frac{1}{ND} \sum_{i=1}^{N} \sum_{d=1}^{D} (\hat{x}_d(t_i) - x_d(t_i))^2 \tag{30}$$

The DGF residual loss supervises the DGF to fit the CTF residual:

$$\mathcal{L}_{\text{dgf}} = \frac{1}{ND} \sum_{i=1}^{N} \sum_{d=1}^{D} \left( \Psi_{\text{DGF}}^{(d)}(t_i) - r_d(t_i) \right)^2 \tag{31}$$

where $r_d(t_i) = x_d(t_i) - \Phi_{\text{CTF}}^{(d)}(t_i)$.

The smoothness loss penalizes rapid CTF variations:

$$\mathcal{L}_{\text{smooth}} = \frac{1}{(N-1)D} \sum_{i=1}^{N-1} \sum_{d=1}^{D} \left( \Phi_{\text{CTF}}^{(d)}(t_{i+1}) - \Phi_{\text{CTF}}^{(d)}(t_i) \right)^2 \tag{32}$$

The sparsity loss encourages parsimonious atom usage:

$$\mathcal{L}_{\text{sparse}} = \lambda_s \left( \bar{z} + \beta \cdot \text{ReLU}(\bar{z} - \rho) \right) \tag{33}$$

where $\bar{z} = \frac{1}{M} \sum_{k=1}^{M} z_k$, target sparsity $\rho = 0.3$, penalty coefficient $\beta = 10$, and $\lambda_s = 0.001$.

### A.7. Hyperparameters and Complexity

Table 6 summarizes all hyperparameters used in our experiments.

*Table 6.* Default hyperparameters.

| Parameter | Symbol | Value | Parameter | Symbol | Value |
|---|---|---|---|---|---|
| Number of frequencies | $K$ | 16 | DGF loss weight | $\lambda_r$ | 0.1 |
| Frequency cutoff | $f_{\max}$ | 8.0 Hz | Smoothness weight | $\lambda_m$ | 0.001 |
| Hidden dimension | $H$ | 64 | Base scale | $\sigma_{\text{base}}$ | 0.1 |
| Number of atoms | $M$ | 16 | Scale floor | $\epsilon$ | 0.02 |
| Gate temperature | $\alpha$ | 5 | Warmup ratio | $e_{\text{warm}}/E$ | 0.3 |
| Target sparsity | $\rho$ | 0.3 | Learning rate | - | $10^{-3}$ |
| Sparsity penalty | $\beta$ | 10 | Weight decay | - | $10^{-4}$ |
| Sparsity weight | $\lambda_s$ | 0.001 | Gradient clip norm | - | 1.0 |

The model contains approximately 35,000 parameters for typical configurations with $D = 7$ variables, $K = 16$ frequencies, $M = 16$ atoms, and $H = 64$ hidden dimension. The parameter count scales as $\mathcal{O}(KH + DH + H^2 + MD + MH)$, dominated by the hidden layer transformations.

The computational complexity per forward pass consists of three main components: (1) FFT-based low-pass filtering requires $\mathcal{O}(ND \log N)$ operations for transforming and filtering the input signal; (2) network forward passes through the CTF and DGF encoders require $\mathcal{O}(N(KH + DH + H^2))$ operations; (3) Gabor atom evaluation requires $\mathcal{O}(NMD)$ operations to compute the dictionary reconstruction. The overall complexity is $\mathcal{O}(ND \log N + NKH + NDH + NH^2 + NMD)$, which is log-linear in sequence length $N$ and comparable to standard sequence models.

Memory usage scales as $\mathcal{O}(ND + NH + MD)$ for storing the input signal, intermediate activations, and atom parameters. For typical configurations ($N = 512$, $D = 7$, $H = 64$, $M = 16$), peak memory consumption is approximately 2-3 MB per sample, enabling efficient batch processing.

Training converges within 300-500 epochs for reconstruction tasks and 500-1000 epochs for interpolation tasks. On a single NVIDIA RTX 3090 GPU, training takes approximately 2-5 minutes for reconstruction (batch size 32, 300 epochs) and 10-30 minutes for interpolation (per-sample optimization, 500 epochs per sample). Inference is efficient, requiring only 1-2 ms per query for continuous-time reconstruction at arbitrary temporal resolutions.

## B. Detailed Experimental Settings

### B.1. Datasets

We evaluate DualTimesField on nine long-horizon time series benchmarks and six irregularly sampled datasets. As illustrated in Figure 4, these benchmarks exhibit diverse temporal patterns. Specifically, Electricity and Traffic show strong seasonality, ILI and Exchange demonstrate complex trends, while Weather is characterized by sparse spikes. Table 7 summarizes the key statistics of all datasets used in our experiments.

The **ETT** (Electricity Transformer Temperature) datasets (Zhou et al., 2021) include hourly and 15-minute measurements from power transformers. **Electricity** (Trindade, 2015), **Exchange** (Lai et al., 2018), **Traffic** (Cuturi, 2011), and **Weather** (Max Planck Institute for Biogeochemistry) are widely used benchmarks spanning energy, finance, transportation, and climate domains. **ILI** (Influenza-Like Illness) (Centers for Disease Control and Prevention, 2025) tracks epidemiological data from the CDC.

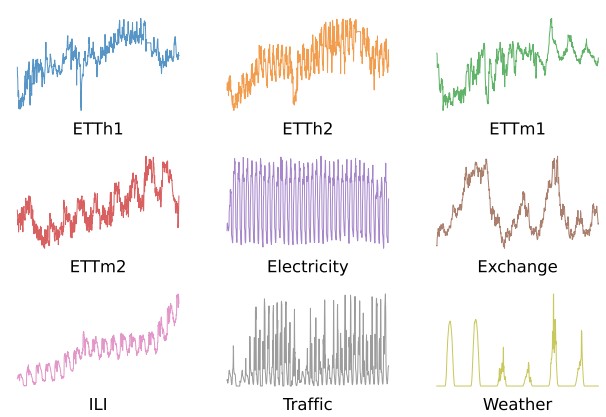

*Figure 4.* Visualization of representative time series from the nine benchmark datasets used in our evaluation.

For irregularly sampled data, **PhysioNet** (Silva et al., 2012) contains multivariate ICU records with missing and asynchronous observations, and **USHCN** (Menne et al., 2016) provides long-term climate measurements with irregular observation masks. **Human Activity** (Shukla & Marlin, 2021) records 3D body-sensor positions with asynchronous timestamps across multiple subjects. We additionally include **EPA-Air**, **ClusterTrace**, and **FN-SPID** from the Time-IMM benchmark (Chang et al., 2025), covering air quality monitoring, cloud-computing cluster traces, and financial news with stock prices, respectively. Within the Time-IMM taxonomy, ClusterTrace and FNSPID exemplify constraint-based irregularity via resource-aware collection and operational-window sampling respectively, while EPA-Air exemplifies artifact-based irregularity via multi-source sensor asynchrony. All datasets follow standard train-test splits from prior work.

### B.2. Baselines

We compare DualTimesField against multiple baseline methods across different paradigms. Table 8 summarizes the key characteristics and limitations of each baseline. For reconstruction, we evaluate against implicit neural representations (SIREN, WIRE, Fourier Features), deep time series models (N-BEATS, PatchTST, iTransformer, TimesNet), and classical dimensionality reduction (PCA). For irregularly sampled time series interpolation, we compare against RNN-based methods (RNN, RNN-VAE, ODE-RNN, GRU-D) and advanced continuous-time models (Latent ODE, SAITS, CRU, LS4). Further details concerning the baselines are shown in Table 8.

*Table 7.* Statistics of the benchmark datasets.

| Dataset | Domain | Length | Variables | Frequency | Train | Test |
|---|---|---|---|---|---|---|
| ETTh1 | Power | 17,420 | 7 | Hourly | 12,000 | 3,420 |
| ETTh2 | Power | 17,420 | 7 | Hourly | 12,000 | 3,420 |
| ETTm1 | Power | 69,680 | 7 | 15-min | 48,000 | 11,520 |
| ETTm2 | Power | 69,680 | 7 | 15-min | 48,000 | 11,520 |
| Electricity | Energy | 26,304 | 321 | Hourly | 18,413 | 5,261 |
| Exchange | Finance | 7,588 | 8 | Daily | 5,311 | 1,519 |
| Traffic | Transport | 17,544 | 862 | Hourly | 12,280 | 3,509 |
| Weather | Climate | 52,696 | 21 | 10-min | 36,887 | 10,540 |
| ILI | Health | 966 | 7 | Weekly | 676 | 193 |
| PhysioNet | Medical | Variable | 37 | Irregular | 7,314 | 915 |
| USHCN | Climate | Variable | 5 | Irregular | 1,114 | 279 |
| Human Activity | Activity | Variable | 12 | Irregular | 5,243 | 1,311 |
| EPA-Air | Air Quality | Variable | 4 | Irregular | 29,731 | 9,910 |
| ClusterTrace | Cloud Comp. | Variable | 11 | Irregular | 41,400 | 13,800 |
| FNSPID | Finance | Variable | 6 | Irregular | 125,812 | 41,938 |

*Table 8.* Comparison of baseline methods.

| Category | Method | Approach | Limitation |
|---|---|---|---|
| *Reconstruction* | | | |
| INR | SIREN (Sitzmann et al., 2020) | Periodic activations | Spectral bias |
| | WIRE (Saragadam et al., 2023) | Complex Gabor wavelets | Complex computation |
| | Fourier (Tancik et al., 2020) | Random Fourier features | Fixed frequencies |
| Deep Models | N-BEATS (Oreshkin et al., 2020) | Interpretable decomposition | Discrete-time only |
| | PatchTST (Nie et al., 2023) | Patch-based Transformer | Channel independence |
| | iTransformer (Liu et al., 2024) | Inverted attention | Coarse time tokens |
| | TimesNet (Wu et al., 2023) | Multi-scale 2D patterns | High memory cost |
| Classical | PCA (Jolliffe & Cadima, 2016) | Linear dimensionality reduction | Linear manifolds |
| *Irregularly Sampled Time Series* | | | |
| RNN-based | RNN (Rumelhart et al., 1985) | Standard recurrence | Regular intervals |
| | RNN-VAE (Chung et al., 2014) | Variational inference | Posterior collapse |
| | ODE-RNN (Rubanova et al., 2019) | Neural ODE dynamics | Expensive solvers |
| | GRU-D (Che et al., 2018) | Time-aware decay | Hand-designed |
| Advanced | Latent ODE (Chen et al., 2018) | Continuous latent space | High complexity |
| | SAITS (Du et al., 2023) | Masked self-attention | Imputation-oriented |
| | CRU (Schirmer et al., 2022) | Continuous recurrence | Gaussian state SDE |
| | LS4 (Zhou et al., 2023) | State space models | Sensitive tuning |

## B.3. Evaluation Metrics

We employ two complementary metrics to assess representation quality across reconstruction fidelity.

**Mean Squared Error (MSE)** measures the average squared difference between predicted and ground-truth values:

$$\text{MSE} = \frac{1}{ND} \sum_{i=1}^{N} \sum_{d=1}^{D} (\hat{x}_{i,d} - x_{i,d})^2 \tag{34}$$

where $N$ is the number of time points, $D$ is the number of variables, $\hat{x}_{i,d}$ is the predicted value, and $x_{i,d}$ is the ground truth.

**Mean Absolute Error (MAE)** measures the average absolute difference:

$$\text{MAE} = \frac{1}{ND} \sum_{i=1}^{N} \sum_{d=1}^{D} |\hat{x}_{i,d} - x_{i,d}| \tag{35}$$

In our implementation, both metrics are computed batch-wise on the test set and averaged across all batches to obtain the final evaluation scores.

## C. Comprehensive Reconstruction Analysis

This section provides extended visual analysis of the reconstruction results reported in Table 1, offering deeper insights into the performance characteristics of DualTimesField across different datasets and method categories.

### C.1. Reconstruction Fidelity Analysis

Figure 5a presents a bubble chart visualization of reconstruction fidelity across all methods and datasets using two complementary error metrics. Each bubble represents a method-dataset combination, with position indicating MAE (x-axis) and MSE (y-axis) on logarithmic scales. DualTimesField (star markers) consistently occupies the optimal lower-left region, achieving both low mean absolute error and low mean squared error simultaneously. In contrast, INR baselines (SIREN (Sitzmann et al., 2020), WIRE (Saragadam et al., 2023), Fourier (Tancik et al., 2020)) cluster in the upper-right region with MSE and MAE values approaching or exceeding 1.0, indicating severe spectral bias that affects both metrics uniformly. Deep learning methods exhibit a clear progression toward better fidelity: N-BEATS (Oreshkin et al., 2020) and PatchTST (Nie et al., 2023) occupy intermediate positions in the bubble landscape, while iTransformer (Liu et al., 2024) and TimesNet (Wu et al., 2023) reach more competitive lower-error regions, yet all still exhibit substantially higher errors than DualTimesField across datasets. The tight clustering of DualTimesField in the lower-left corner demonstrates that explicit trend-event decomposition consistently delivers superior reconstruction fidelity on both error measures.

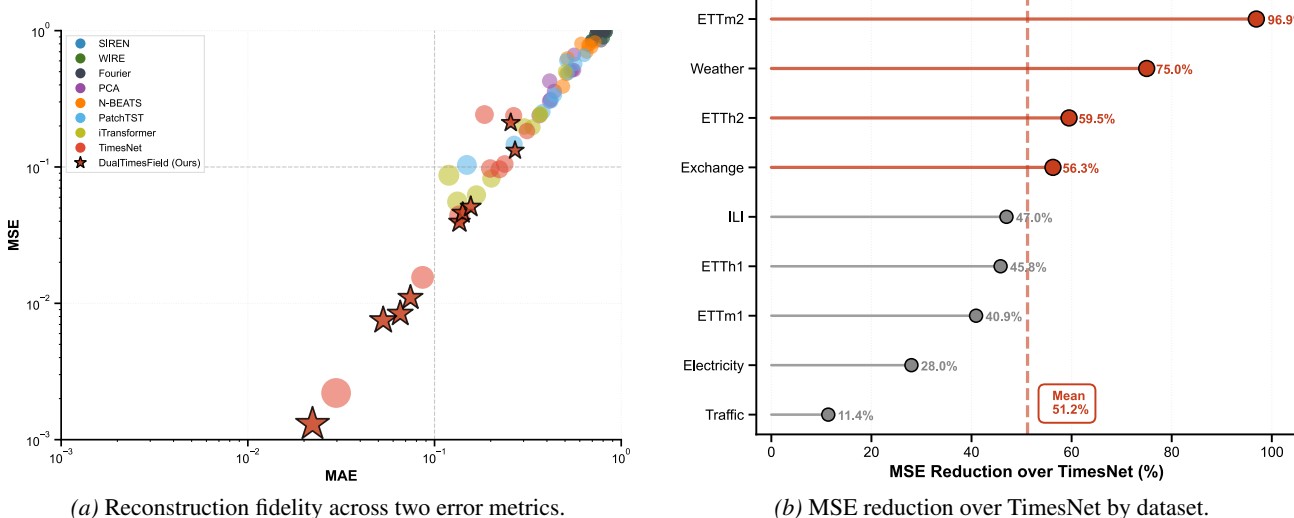

*(a)* Reconstruction fidelity across two error metrics.  *(b)* MSE reduction over TimesNet by dataset.

*Figure 5.* (a) Reconstruction fidelity across two error metrics. Bubbles show MAE (x) vs. MSE (y) on log scales. DualTimesField (stars) consistently achieves the lowest errors on both metrics. (b) Dataset-wise MSE reduction over TimesNet. Red and gray denote improvements above and below 50%, respectively. The dashed line indicates the average reduction (51.2%).

### C.2. Relative Improvement Analysis

Figure 5b shows that DualTimesField consistently outperforms TimesNet across all datasets, achieving an average MSE reduction of 51.2%. The largest gains are observed on ETTm2 (96.9%), Weather (75.0%), and ETTh2 (59.5%). Notably, datasets exhibiting strong periodic patterns benefit the most, directly reflecting the effectiveness of the proposed frequency-domain modeling in capturing long-range temporal regularities. Even the smallest improvement, observed on the Traffic

dataset (11.4%), represents a meaningful reduction in reconstruction error. These consistent improvements across diverse domains validate that the dual-field decomposition provides systematic benefits rather than dataset-specific advantages.

## C.3. Performance Distribution Analysis

Figure 6 presents box plots showing the distribution of MSE across datasets for each method. DualTimesField achieves the lowest median MSE and the smallest interquartile range, indicating consistently superior reconstruction quality across diverse domains. INR methods exhibit tight distributions but at uniformly high error levels, while PatchTST, iTransformer, and TimesNet show progressively lower medians with substantial variance and occasional outliers. The consistent performance of DualTimesField across all nine datasets demonstrates that the dual-field decomposition provides robust improvements rather than dataset-specific advantages, with error distributions centered well below all baseline methods.

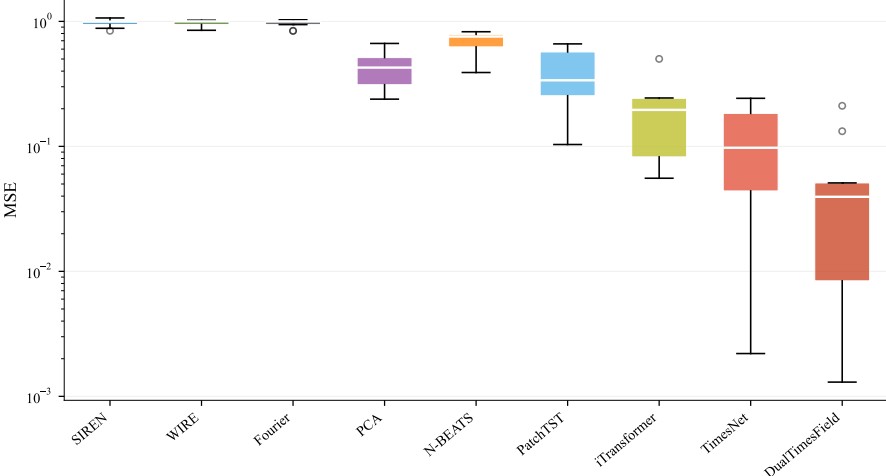

*Figure 6.* Box plot comparison of MSE distributions across datasets. DualTimesField achieves the lowest median MSE with minimal variance across all benchmarks.

## C.4. Average Performance Summary

Figure 7 summarizes average performance across all datasets for each method. The two panels show average MSE and average MAE respectively. DualTimesField achieves the lowest average MSE (0.056) and MAE (0.131), substantially outperforming all baselines. The dual-field architecture demonstrates consistent superiority on both error metrics, with INR methods exhibiting near-unity errors due to spectral bias, while deep learning baselines show moderate but still substantially higher errors than DualTimesField. This summary confirms that DualTimesField's advantages are consistent rather than dataset-specific, delivering robust improvements across diverse domains.

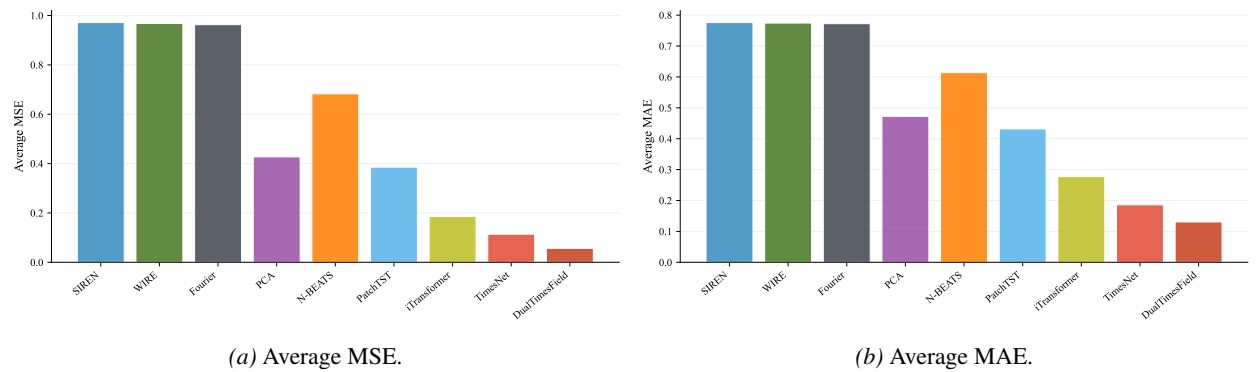

*(a)* Average MSE.          *(b)* Average MAE.

*Figure 7.* Average performance summary across all datasets. (a) Reconstruction error (MSE) and (b) Absolute error (MAE). DualTimesField achieves the best performance on both metrics.

## D. Decomposition and Reconstruction Visualizations

This section presents qualitative analysis of the learned dual-field decomposition and reconstruction quality on four representative datasets spanning diverse temporal characteristics. In decomposition figures, the original signal (blue) is separated into the CTF trend component (green) and the DGF event component (magenta). In reconstruction figures, the original signal (blue) is overlaid with the reconstructed output (red dashed).

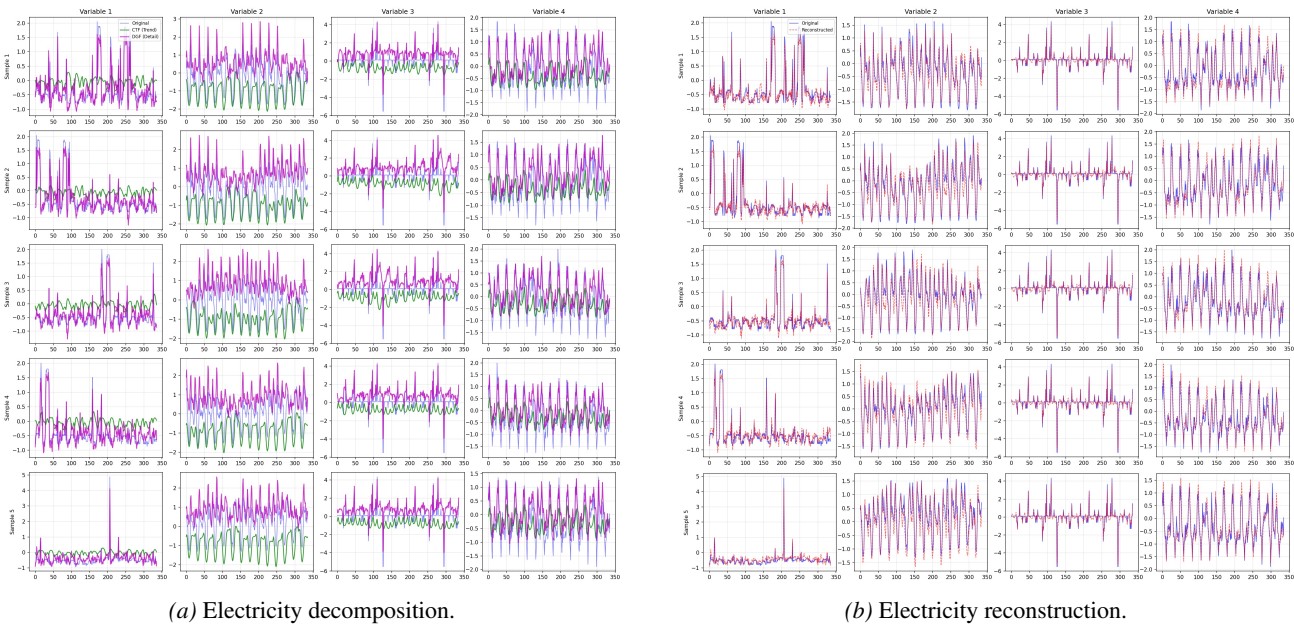

*(a)* Electricity decomposition.                     *(b)* Electricity reconstruction.

*Figure 8.* Decomposition and reconstruction on Electricity (321 variables). DualTimesField preserves fine-grained consumption patterns while effectively separating trends from irregular spikes.

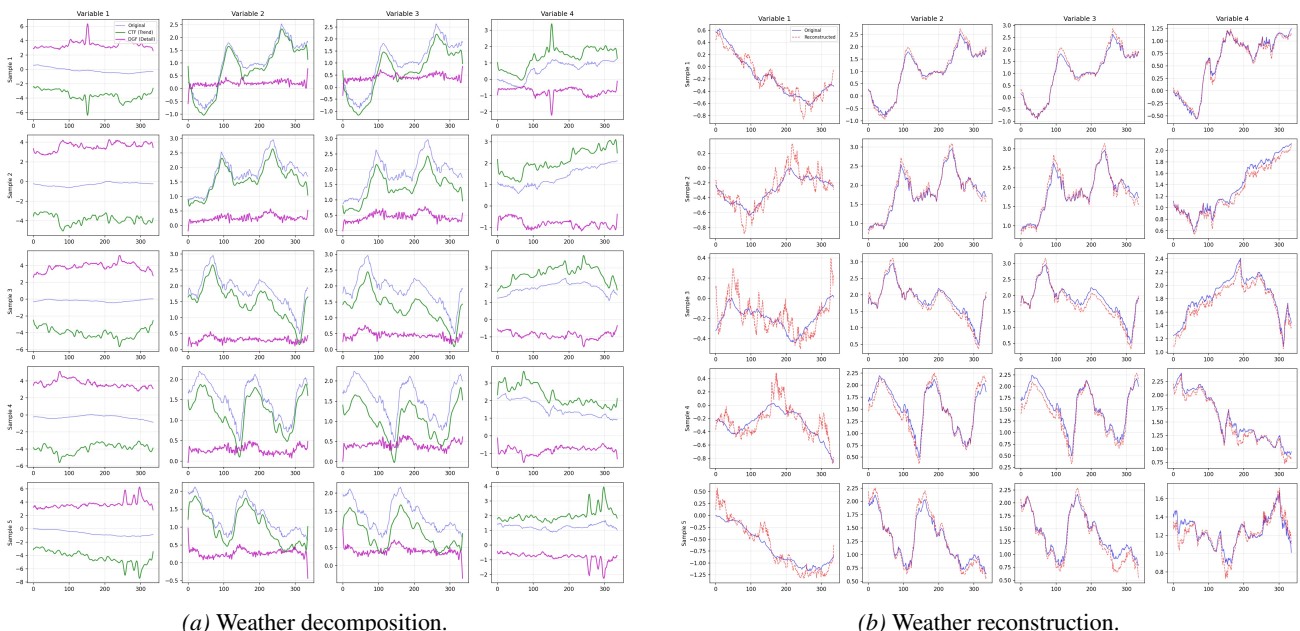

*(a)* Weather decomposition.                         *(b)* Weather reconstruction.

*Figure 9.* Decomposition and reconstruction on Weather. The smooth atmospheric dynamics are primarily captured by the CTF, with minimal DGF activation.

Figure 8 demonstrates performance on the Electricity dataset (321 variables), where the CTF captures dominant daily consumption patterns while the DGF isolates irregular usage spikes, maintaining excellent reconstruction fidelity across

all variables. Figure 9 presents results on Weather (21 variables), where the CTF dominates the representation capturing smooth diurnal variations, with the DGF remaining quiescent except for occasional rapid transitions. Figure 10 illustrates the decomposition and reconstruction on ETTh1, an hourly-sampled energy dataset with 7 variables. The CTF captures smooth underlying temperature dynamics, while the DGF responds to rapid fluctuations with sparse but precise activations at locations of genuine anomalies.

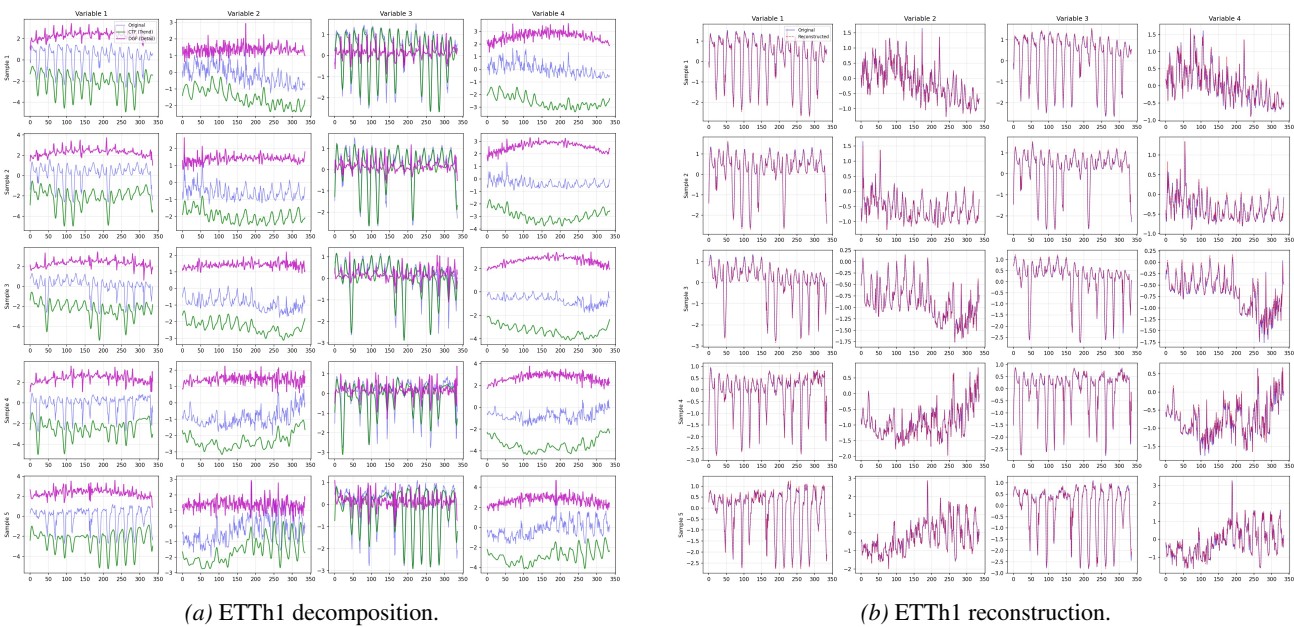

*(a)* ETTh1 decomposition.                    *(b)* ETTh1 reconstruction.

*Figure 10.* Decomposition and reconstruction on ETTh1. The dual-field architecture effectively separates smooth trends from transient events while maintaining high reconstruction fidelity.

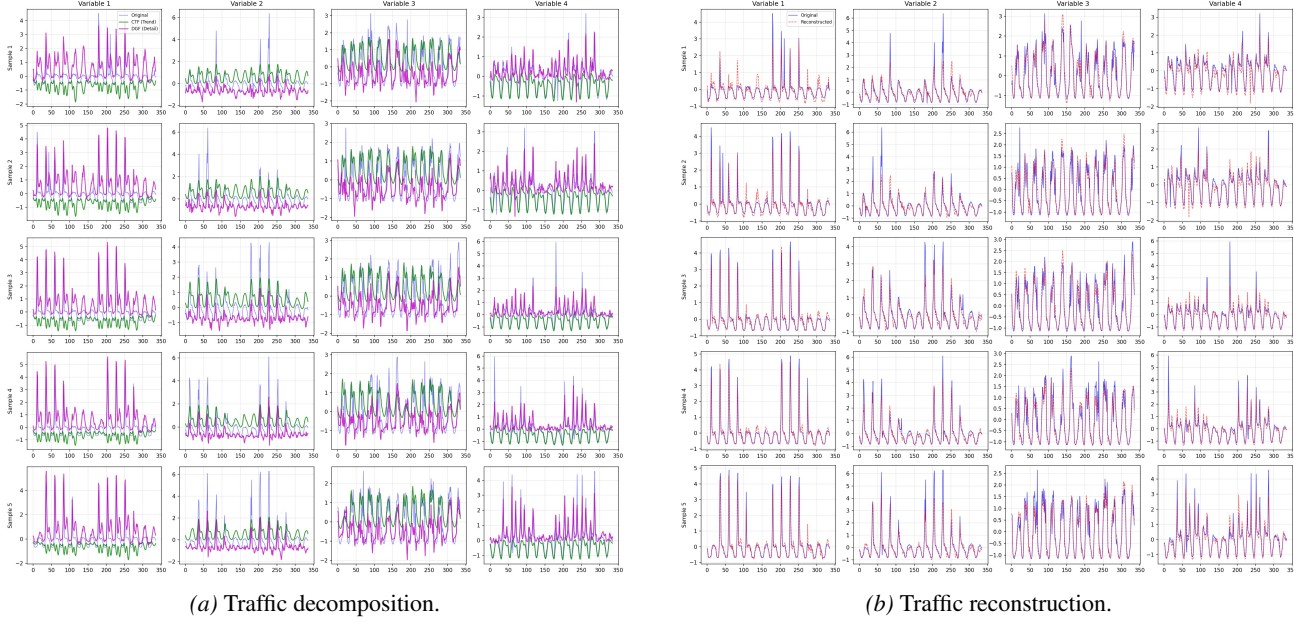

*(a)* Traffic decomposition.                    *(b)* Traffic reconstruction.

*Figure 11.* Decomposition and reconstruction on Traffic (862 sensors). The dual-field architecture efficiently separates regular commute patterns from transient traffic events.

Figure 11 shows results on Traffic (862 sensors), exhibiting the most complex temporal structure with strong daily periodicity superimposed with frequent transient events. DualTimesField maintains high reconstruction quality by efficiently separating regular commute patterns from transient traffic events across all sensor locations.

# E. Interpolation Quality Analysis

We provide qualitative analysis of interpolation performance on PhysioNet (Silva et al., 2012) and USHCN (Menne et al., 2016), two representative datasets from our six-benchmark irregular sampling suite covering medical and climate domains. Figure 12 shows predicted values against ground truth for withheld target observations across a representative subset of 50 time series samples per dataset, where each sample contributes multiple withheld target points as determined by the observation masks, resulting in several thousand prediction-truth pairs. Points closely follow the diagonal line of perfect prediction, with PhysioNet achieving $R^2 = 0.9975$ and MSE of $0.14 \times 10^{-3}$, while USHCN achieves $R^2 = 0.9895$ and MSE of $0.86 \times 10^{-3}$. The error distributions in Figure 13 are tightly concentrated around zero with near-zero mean errors (PhysioNet: $-0.0000$; USHCN: $-0.0012$), indicating unbiased and accurate predictions.

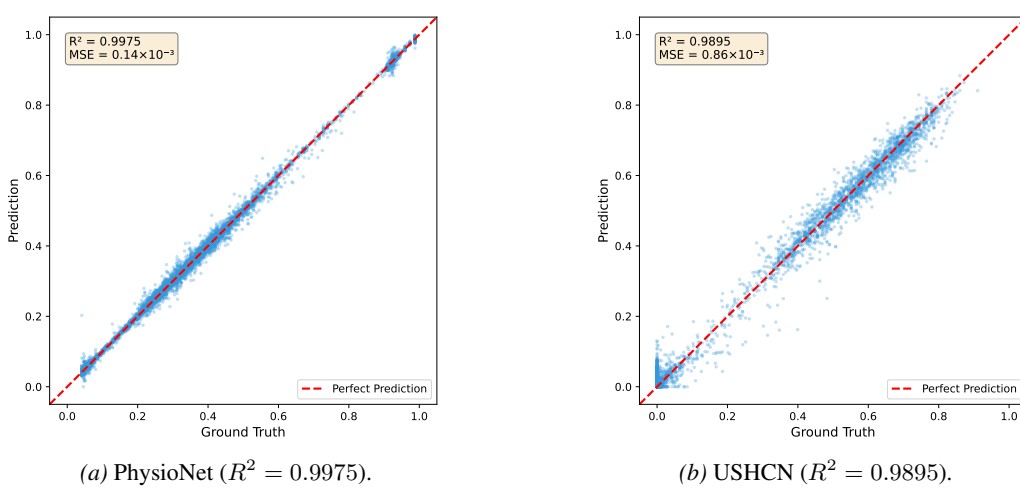

*(a)* PhysioNet ($R^2 = 0.9975$).    *(b)* USHCN ($R^2 = 0.9895$).

*Figure 12.* Prediction vs. ground truth on irregularly sampled datasets.

Note that these visualization metrics aggregate predictions across all withheld points from the 50-sample subset, while Table 2 reports per-sample averaged MSE over the full test set (PhysioNet: 915 samples with MSE $0.30 \times 10^{-3}$; USHCN: 279 samples with MSE $0.17 \times 10^{-3}$). The different aggregation strategies explain the numerical differences, but both evaluation protocols consistently demonstrate that DualTimesField effectively handles irregular sampling patterns and provides reliable continuous-time interpolation across medical and climate monitoring scenarios.

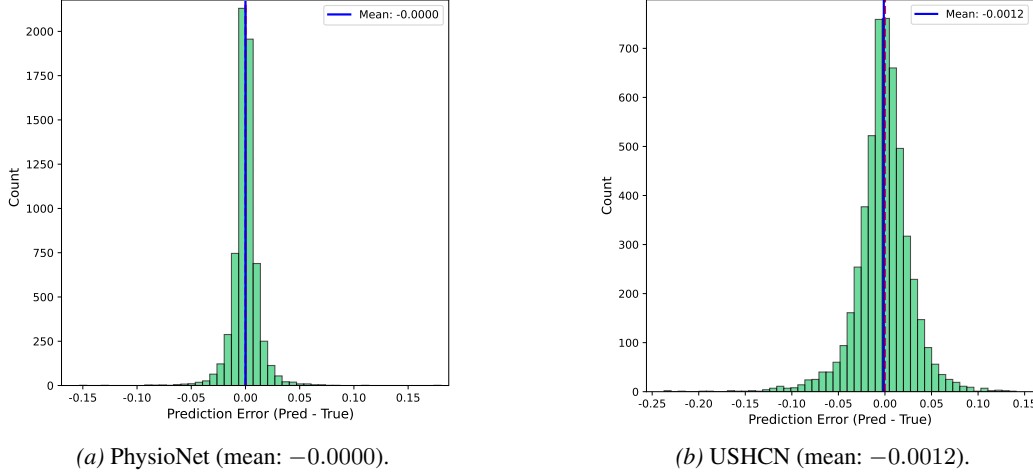

*(a)* PhysioNet (mean: $-0.0000$).    *(b)* USHCN (mean: $-0.0012$).

*Figure 13.* Error distributions for interpolation predictions.

## F. Limitations

Several limitations of DualTimesField warrant discussion. First, the optimization process requires iterative training that scales with dataset size, limiting applicability in time-critical deployment scenarios. Second, the framework involves multiple hyperparameters (number of frequencies, atoms, frequency cutoff, loss weights) that may require dataset-specific tuning; while we provide default settings that work well across benchmarks, systematic sensitivity analysis remains an avenue for future investigation. Third, performance varies across data characteristics: on USHCN, structured state-space models achieve superior interpolation, with LS4 (Zhou et al., 2023) and CRU (Schirmer et al., 2022) attaining MSE of $0.06 \times 10^{-3}$ and $0.16 \times 10^{-3}$ respectively, compared with $0.17 \times 10^{-3}$ for DualTimesField. This suggests that climate data dominated by smooth periodic patterns may benefit less from explicit trend-event decomposition. Finally, the current implementation assumes regularly sampled training data for reconstruction tasks, though the continuous-time formulation naturally supports irregular sampling as demonstrated in interpolation experiments.

## G. Future Work

The dual-field framework opens several promising research directions. Architecturally, extending to hierarchical multi-scale representations with multiple fields at different temporal resolutions could capture richer nested dynamics beyond two-level decomposition. Methodologically, incorporating learnable frequency selection or adaptive atom placement could reduce hyperparameter sensitivity. From an application perspective, investigating how learned trend-event decompositions transfer to downstream tasks—forecasting, classification, anomaly detection—would demonstrate practical value, where explicit separation may provide both performance gains and interpretable diagnostics. Additionally, exploring the framework's potential for streaming or online learning scenarios, where the continuous representation could enable efficient incremental updates, presents an interesting direction for real-time applications.

