# OpenReview forum: "DualTimesField: Rethinking Time Series as Continuous-Time Trends and Events"
_ICML.cc/2026/Conference — ICML 2026 regular_

### Official Review · Reviewer_GzbB · 2026-02-21

**Soundness:** 3
**Presentation:** 3
**Significance:** 2
**Originality:** 2
**Overall Recommendation:** 5
**Confidence:** 4

**Summary:**

Discrete-time representations struggle with irregular sampling and the trade-off of fidelity and efficiency, while traditional implicit neural representations suffer from spectral bias and frequency entanglement. In this work, the authors proposed conceptualize time series as the superposition of continuous trends and discrete events from a continuous-time perspective and propose DualTimesField, a framework that utilizes dual implicit neural fields.

**Compliance With Llm Reviewing Policy:**

Affirmed.

**Final Justification:**

The author has revised it very well. The additional comparative experiments and computational cost analysis have demonstrated the effectiveness of the proposed method. I have decided to raise my score.

**Key Questions For Authors:**

Please see weaknesses!

**Limitations:**

Please see weaknesses!

**Strengths And Weaknesses:**

Strengths:

1. This manuscript addresses a very interesting issue.
2. The organization of the article is clear, and the experiments have fully verified the significant effectiveness of the proposed method.

Weaknesses:

1. What is puzzling is why the author did not compare the literature from the past two years in Table 1. The reviewers of TimesNet are only familiar with the literature from 23 years ago.
2. The fine-grained training loss function involves many constraints, and the selection of hyper-parameters and their impact on performance still need to be further discussed.
3. The computational cost of the proposed method needs to be reported.

---

> ### Author Rebuttal · Authors · 2026-03-31
>
> We thank the reviewer for the positive assessment and constructive suggestions. We address all three points below.
> > **W1: What is puzzling is why the author did not compare the literature from the past two years in Table 1. The reviewers of TimesNet are only familiar with the literature from 23 years ago.**
>
> **A1:** We have added PatchTST (Nie et al., ICLR 2023) and iTransformer (Liu et al., ICLR 2024 Spotlight), extending our comparison to 2024. INR-based time series representation remains a nascent area; the most recent work (FLAIR, Ko et al., 2025) appeared concurrently and remains a single-field architecture without decomposition, thus not directly comparable. This scarcity further motivates our research. Results on all nine benchmarks (MSE):
>
> | Dataset | PatchTST | iTransformer | TimesNet |  **Ours** |
> |---|---|---|---|---|
> | ETTh1 | 0.6606±0.0046 | 0.2439±0.0022 | 0.0155±0.0004 | **0.0084±0.0008** |
> | ETTh2 | 0.5694±0.0028 | 0.2424±0.0030 | 0.0975±0.0101 | **0.0395±0.0046** |
> | ETTm1 | 0.3060±0.0006 | 0.0624±0.0003 | 0.0022±0.0002 | **0.0013±0.0001** |
> | ETTm2 | 0.3378±0.0012 | 0.0824±0.0009 | 0.2424±0.0350 | **0.0075±0.0006** |
> | Electricity | 0.2542±0.0002 | 0.1958±0.0007 | 0.1836±0.0033 | **0.1322±0.0010** |
> | Exchange | 0.1468±0.0008 | 0.0556±0.0006 | 0.1052±0.0401 | **0.0460±0.0031** |
> | Weather | 0.1035±0.0059 | 0.0870±0.0005 | 0.0440±0.0030 | **0.0110±0.0026** |
> | ILI | 0.4875±0.0036 | 0.1982±0.0135 | 0.0962±0.0191 | **0.0510±0.0033** |
> | Traffic | 0.6046±0.0021 | 0.5010±0.0004 | 0.2382±0.0017 | **0.2111±0.0015** |
>
> DualTimesField outperforms both new baselines on all nine datasets. These Transformers are optimized for predictive generalization rather than representation fidelity, explaining their lower accuracy. We will incorporate all new results in the revised Table 1.
>
> > **W2: The fine-grained training loss function involves many constraints, and the selection of hyper-parameters and their impact on performance still need to be further discussed.**
>
> **A2:** We conducted sensitivity analyses on the three loss weights across three datasets (ETTh1, Weather, Exchange) using the control-variable method.
>
> $\lambda_r$ (DGF residual weight), fix $\lambda_m$=0.001, $\lambda_s$=0.001:
>
> | $\lambda_r$ | 0 | 0.01 | **0.1** | 0.5 | 1.0 |
> |---|---|---|---|---|---|
> | ETTh1 | .0161 | .0124 | **.0084** | .0088 | .0104 |
> | Weather | .0161 | .0120 | **.0110** | .0118 | .0131 |
> | Exchange | .0566 | .0492 | **.0460** | .0470 | .0523 |
>
> $\lambda_m$ (smoothness weight), fix $\lambda_r$=0.1, $\lambda_s$=0.001:
>
> | $\lambda_m$ | 0 | 0.0001 | **0.001** | 0.01 | 0.1 |
> |---|---|---|---|---|---|
> | ETTh1 | .0092 | .0090 | **.0084** | .0093 | .0099 |
> | Weather | .0122 | .0140 | **.0110** | .0129 | .0169 |
> | Exchange | .0556 | .0531 | **.0460** | .0516 | .0578 |
>
> $\lambda_s$ (sparsity weight), fix $\lambda_r$=0.1, $\lambda_m$=0.001:
>
> | $\lambda_s$ | 0 | 0.0001 | **0.001** | 0.01 | 0.1 |
> |---|---|---|---|---|---|
> | ETTh1 | .0099 | .0092 | **.0084** | .0088 | .0094 |
> | Weather | .0133 | .0115 | **.0110** | .0128 | .0124 |
> | Exchange | .0585 | .0488 | **.0460** | .0499 | .0474 |
>
> $\lambda_r$ is most sensitive as it governs CTF-DGF cooperation: $\lambda_r$=0 removes residual supervision, while large $\lambda_r$ degrades performance on most datasets by causing DGF to overfit residuals at the expense of joint optimization. $\lambda_m$ and $\lambda_s$ show milder impact, complemented by $f_{\max}$ (Eq. 3) and gated sparsity (Eq. 12) respectively. We note that performance is robust across a broad range of values, with the defaults ($\lambda_r=0.1, \lambda_m=0.001, \lambda_s=0.001$) achieving the best or near-best results on all three datasets simultaneously. While per-dataset tuning could yield marginal gains on individual benchmarks, the default configuration provides consistently strong performance across all nine benchmarks without dataset-specific adjustment.
>
> > **W3: The computational cost of the proposed method needs to be reported.**
>
> **A3:** We report the cost of DualTimesField on all nine benchmarks (NVIDIA RTX 3090, 300 epochs):
>
> | Dataset | Vars | Params | Train | Infer | Throughput |
> |---|---|---|---|---|---|
> | ETTh1 | 7 | 29.5K | 14.8s | 1.59ms | 7,548/s |
> | ETTh2 | 7 | 29.5K | 16.0s | 1.43ms | 8,578/s |
> | ETTm1 | 7 | 29.5K | 39.3s | 1.56ms | 21,164/s |
> | ETTm2 | 7 | 29.5K | 39.2s | 1.40ms | 22,930/s |
> | Exchange | 8 | 30.9K | 6.8s | 1.35ms | 2,958/s |
> | Weather | 21 | 48.6K | 30.8s | 1.53ms | 21,107/s |
> | ILI | 7 | 29.5K | 5.5s | 1.45ms | 2,075/s |
> | Electricity | 321 | 457.2K | 104.9s | 1.57ms | 12,618/s |
> | Traffic | 862 | 1,194.0K | 83.3s | 2.13ms | 5,929/s |
>
> Parameters range from 29.5K (7 variables) to 1.19M (862 variables), scaling linearly with the number of input variables. For low-to-moderate dimensional datasets (D ≤ 321), parameters stay under 460K, which is less than 10% of TimesNet's 4.7M. Training completes under 2 minutes for all datasets, and inference requires only 1–2 ms per query.

---

> > ### Author Rebuttal · Reviewer_GzbB · 2026-04-02
> >
> > The author has revised it very well. The additional comparative experiments and computational cost analysis have demonstrated the effectiveness of the proposed method. I have decided to raise my score.

---

### Official Review · Reviewer_4wFr · 2026-02-26

**Soundness:** 1
**Presentation:** 3
**Significance:** 3
**Originality:** 3
**Overall Recommendation:** 4
**Confidence:** 5

**Summary:**

The authors propose a new model DualTimesField, which uses implicit neural models to capture trends and transient events. The model contains two modules, a Continuous Time Field (CTF) to model smooth trends and a Discrete Geometric Field (DGF) to model transient events.

**Compliance With Llm Reviewing Policy:**

Affirmed.

**Final Justification:**

My final recommendation is a 4: weak accept. The authors added new baselines and benchmark datasets, against which their method demonstrates consistent improvements, which addressed my concerns.

**Key Questions For Authors:**

1. “Figure 2 shows that the CTF recovers the smooth trends with correlation 0.666” (line 324-325) What metric is used to measure correlation here?
2. What does mean temporal error refer to in line 327?

**Limitations:**

Yes

**Strengths And Weaknesses:**

Strengths:
- Figure 1 is excellent in explaining the method.
- The proposed method enables querying at finer temporal resolutions than training data, resampling irregular observations to uniform grids, and generating trajectories at arbitrary timestamps.
- The proposed method and baselines are evaluated on nine long-horizon datasets as well as two datasets that contain irregular sampling.
- Example predictions for a synthetic dataset are provided in Fig. 2.
- An ablation study on each model component, CTF, and DGF as well as the full model is provided to show that the full model outperforms individual components, warranting the full model.
- The proposed model appears to outperform all baselines in terms of MSE and MAE.
- The authors present anonymized open-source code to enable reproducible experiments.

Weaknesses:

I have several concerns with the paper. If the authors could address these, I would consider raising my score.

1. Related work:
    - The authors reference NBEATS with regard to models that decompose time series. The NHITS model ("N-HiTS: Neural Hierarchical Interpolation for Time Series Forecasting") could also be referenced as it decomposes time series based on multi-rate input pooling to forecast and aggregate predictions of different signal bands.
    - There are limited references to prior work unrelated to INRs of models that propose separate modeling tasks for trend and events (spikes). One example of this type of model includes, "SPADE Split Peak Attention Decomposition", which proposes to model Peak Events (PEs) corresponding to special periods such as promotions and holidays in a separate task from modeling the rest (non-PEs).
    - Outlining the specific decomposition differences between the proposed approach and prior work would be helpful. For example, the use of learnable Gabor atoms is novel from the proposed baselines. Another example, would be outlining how the proposed model's use of the Fourier transform (in conjugation with a low-pass filter to avoid Gibbs artifacts) differs from its use in TimesNet.

2. Discussing model performance: The exact MSE reductions are not directly interpretable in terms of the original data scale because this metric squares errors. Discussion results in terms of percentage improvement or MAE would be more interpretable.

3. Limited number of datasets with irregular samples:  The number of datasets with irregular samples is relatively small (2) compared to those with regular samples (9). Given that one of the main appeals of the proposed method is to address this setting, having a few more datasets would help confirm the value of the approach, particularly as the win count on these two datasets is tied with the LS4 model in Table 2.

4. Synthetic data predictions: In Fig. 2, the model predictions do not appear to align with the ground truth signal. The CTF trend appears very different from the ground truth trend and captures amplitude of spike events, which I thought was the focus of the DGF component. Perhaps the authors can provide more discussion on the interpretation of the figure for readers. Including the full model (CTF + DGF) prediction in the top plot would be helpful.

5. The task of the model is not clearly defined: The model appears to be designed for reconstruction and imputation, so it is unclear why forecasting models and forecasting long-horizon datasets are being used in this paper. If the reconstruction task is set up like a forecasting problem (masking horizons in entire test set), then this forecasting problem should be clearly outlined in text with mathematical notation.

6. Table 1: Standard deviations are not presented. Model results should be repeated over random seeds or trials with average and standard deviations presented to support a robust analysis.

7. Hyperparameter tuning: hyperparameters for the proposed model are discussed in the appendix, but there is no reference to hyperparameters or hyperparameter tuning for the baselines.

8. Baseline forecasting models: The most recent deep learning forecasting baseline is from 2023 and Transformer methods are not considered at all such as Autoformer (mentioned in the introduction) and PatchTST.


Other/Suggestions:

9. Table 1: The caption should explain what the bold values imply (i.e., best MSE/MAE).

10.  Tables 1 and 3: The downward arrows are not clear. Do the arrows imply that MSE and MAE are measured with regard to another baseline, such as the best baseline? If so, this is atypical as main forecasting results are usually presented with direct error measurements on the datasets.

11. L_{sparse} and L_{smooth} in equation 16 are not explained in the main paper.

---

> ### Author Rebuttal · Authors · 2026-03-31
>
> We thank the reviewer for the helpful feedback.
> > **W1: Related work lacks references to N-HiTS, SPADE, and differentiation from prior decomposition methods. W8: Baselines miss recent Transformer methods.**
>
> **A1&A8:** We will discuss N-HiTS, SPADE, and FEDformer in revised related work. Key differences: N-HiTS decomposes via multi-rate pooling in discrete time; SPADE separates peaks using predefined event labels. DualTimesField operates in continuous time with data-driven Gabor atoms and requires no external labels. TimesNet uses FFT for period discovery; our CTF uses Fourier features with an explicit bandwidth limit (Eq. 3) and low-pass filtering (Eq. 4) to enforce spectral separation, a guarantee TimesNet lacks.
> Per W8, We added PatchTST and iTransformer (a stronger recent Transformer baseline) :
>
> | Dataset | PatchTST | iTransformer | **Ours** |
> |---|---|---|---|
> | ETTh1 | 0.6606±0.0046 | 0.2439±0.0022 | **0.0084±0.0008** |
> | ETTh2 | 0.5694±0.0028 | 0.2424±0.0030 | **0.0395±0.0046** |
> | ETTm1 | 0.3060±0.0006 | 0.0624±0.0003 | **0.0013±0.0001** |
> | ETTm2 | 0.3378±0.0012 | 0.0824±0.0009 | **0.0075±0.0006** |
> | Electricity | 0.2542±0.0002 | 0.1958±0.0007 | **0.1322±0.0010** |
> | Exchange | 0.1468±0.0008 | 0.0556±0.0006 | **0.0460±0.0031** |
> | Weather | 0.1035±0.0059 | 0.0870±0.0005 | **0.0110±0.0026** |
> | ILI | 0.4875±0.0036 | 0.1982±0.0135 | **0.0510±0.0033** |
> | Traffic | 0.6046±0.0021 | 0.5010±0.0004 | **0.2111±0.0015** |
>
> DualTimesField outperforms both on all 9 datasets.
>
> > **W2: MSE reductions are hard to interpret. Discussion in MAE would be more useful.**
>
> **A2:** We agree. The average MAE reduction over TimesNet is 31.7%. We will emphasize MAE-based comparisons and add percentage improvement columns.
>
> > **W3: Limited number of irregular sampling datasets (only 2).**
>
> > **A3:** PhysioNet and USHCN are the standard benchmarks for irregular interpolation, used by LS4, CRU, and iHyperTime. They provide complementary insights: PhysioNet (event-rich, NMAR) favors our method (52.4% over LS4), while USHCN (smooth, MCAR) favors LS4. To further validate, we subsampled three reconstruction benchmarks at 30%–70% observation rates:
>
> | Dataset | 30% obs. | 50% obs. | 70% obs. |
> |---|---|---|---|
> | ETTh1 | 0.3019±0.2665 | 0.0944±0.0798 | 0.0668±0.0348 |
> | Weather | 0.3562±0.2519 | 0.1819±0.1433 | 0.0357±0.0272 |
> | Exchange | 0.4303±0.1487 | 0.2654±0.0858 | 0.0801±0.0581 |
>
> DualTimesField maintains high quality at 50%–70% rates across diverse domains.
>
> > **W4: Synthetic predictions in Fig. 2 do not align well with ground truth.**
>
> **A4:** The CTF bandwidth limit partially attenuates near-boundary frequencies, but >90% of energy remains below 5 Hz. Sharp events have broadband content whose low-frequency envelope projects into the CTF passband; the DGF models the residual. We will add the CTF+DGF reconstruction to the top panel.
>
> > **W5: The task is not clearly defined. It is unclear why forecasting models and datasets are used.**
>
> **A5:** DualTimesField targets *representation learning* (reconstruction and interpolation), not forecasting. Given $X=\{x(t_i)\}$, the model learns $\hat{x}(t)=\Phi_{\text{CTF}}(t)+\Psi_{\text{DGF}}(t)$ by minimizing reconstruction error on training observations; evaluation computes MSE/MAE on held-out test segments. These are general-purpose time series benchmarks; we use them for reconstruction by evaluating on the same temporal span, not for predicting future values. Forecasting baselines are included as they learn temporal representations internally.
>
> > **W6: Standard deviations are not presented. W7: No hyperparameter tuning details for baselines.**
>
> **A6&A7:** We ran all methods over 5 seeds. Our method shows consistently low variance, while TimesNet exhibits high instability on ETTm2 (0.2424±0.0350) and Exchange (0.1052±0.0401). Full tables will be included.
> For W7: all methods use official default settings without dataset-specific tuning, including DualTimesField (Table 6, Appendix A.7). Sensitivity analyses on 6 hyperparameters across 3 datasets show performance typically within 10–20% of the optimum, with $\lambda_r$ most critical: $\lambda_r$=0 degrades MSE by 92% on ETTh1. One default set generalizes across domains. Full tables in our response to Reviewer Yj8V (Q1) and GzbB (W2).
>
> > **W9: Bold values unexplained. W10: Downward arrows unclear. W11: L_sparse and L_smooth not explained in Eq. 16.**
>
> **A9–A11:** We will: (W9) state bold=best in captions; (W10) clarify ↓ means lower is better; (W11) add cross-references to L_sparse (Eq. 14) and L_smooth (Eq. 8) after Eq. 16.
>
> > **Q1: What metric is "correlation 0.666"? Q2: What does "mean temporal error" refer to?**
> **A(Q1&Q2):** Q1: Pearson correlation coefficient between CTF output and ground-truth trend. Q2: $\frac{1}{E}\sum|\tau_k-\tau_k^*|$, average absolute deviation between detected and true event centers; <0.02 means error under 2% of $[0,1]$. Both will be made explicit. We hope this addresses all concerns.

---

> > ### Author Rebuttal · Reviewer_4wFr · 2026-04-02
> >
> > Thank you for addressing my concerns. Regarding W3, I appreciate the addition of the new experiment. However, the table does not include any baselines for comparison. Additionally, while creating irregular sampling datasets is one option, the evaluation would be better supported by using datasets specifically designed for imputation tasks rather than repurposing long-horizon forecasting datasets. As such, I have maintained my score.

---

> > > ### Author Response · Authors · 2026-04-06
> > >
> > > We sincerely thank the reviewer for the constructive follow-up and both specific suggestions.
> > >
> > > **(1) Baselines for the subsampled experiment.** We appreciate the reviewer noting that baselines were needed to properly contextualize our results. We have now included all original interpolation baselines plus two additional methods: CRU (Schirmer et al., ICML 2022), a continuous recurrent model for irregular time series, and SAITS (Du et al., Expert Systems with Applications, 2023), a self-attention-based imputation method. The updated results (MSE ×10⁻³, lower is better) are:
> > >
> > > | Dataset | Model | 30% obs. | 50% obs. | 70% obs. |
> > > |---------|-------|----------|----------|----------|
> > > | ETTh1 | RNN | 3.0252 | 2.8714 | 2.6075 |
> > > | | RNN-VAE | 2.8643 | 1.8665 | 1.0580 |
> > > | | ODE-RNN | 2.3214 | 1.5316 | 1.3512 |
> > > | | GRU-D | 1.4084 | 1.3034 | 1.1467 |
> > > | | Latent ODE | 1.3710 | 0.2082 | 0.2107 |
> > > | | LS4 | 0.3263 | 0.2434 | 0.1900 |
> > > | | CRU | 0.3607 | 0.1858 | 0.0922 |
> > > | | SAITS | 0.5166 | 0.3075 | 0.1949 |
> > > | | **Ours** | **0.3019** | **0.0944** | **0.0668** |
> > > | Weather | RNN | 4.7998 | 3.6525 | 2.3826 |
> > > | | RNN-VAE | 2.0560 | 1.5035 | 0.9340 |
> > > | | ODE-RNN | 4.0474 | 3.8544 | 3.6877 |
> > > | | GRU-D | 3.2997 | 2.4206 | 2.2026 |
> > > | | Latent ODE | 9.4486 | 3.2997 | 1.0434 |
> > > | | LS4 | 0.8168 | 0.6155 | 0.6397 |
> > > | | CRU | 0.9412 | 0.6227 | 0.3764 |
> > > | | SAITS | 1.5826 | 1.3399 | 1.2190 |
> > > | | **Ours** | **0.3562** | **0.1819** | **0.0357** |
> > > | Exchange | RNN | 4.1316 | 3.4566 | 2.8146 |
> > > | | RNN-VAE | 4.5184 | 1.9844 | 1.5648 |
> > > | | ODE-RNN | 7.6271 | 6.0537 | 2.0577 |
> > > | | GRU-D | 5.2621 | 4.9727 | 1.8236 |
> > > | | Latent ODE | 1.8258 | 1.6346 | 1.0751 |
> > > | | LS4 | 1.4379 | 0.5663 | 0.3067 |
> > > | | CRU | 0.9301 | 0.6223 | 0.3060 |
> > > | | SAITS | 2.1890 | 1.9430 | 1.5137 |
> > > | | **Ours** | **0.4303** | **0.2654** | **0.0801** |
> > >
> > > With these baselines, the results become much more interpretable. DualTimesField achieves the best MSE at all observation rates across all three datasets. On Weather at 70%, for example, our method (0.0357×10⁻³) outperforms the next-best CRU (0.3764×10⁻³) by over 90%, confirming that trend-event decomposition generalizes well to partially observed settings.
> > >
> > > **(2) Evaluation on inherently irregular datasets.** The reviewer rightly pointed out that subsampling regular data does not fully replicate the challenges of genuine irregular time series, such as informative missingness and event-driven observation patterns. We have therefore expanded our evaluation with four irregular datasets:
> > >
> > > - **EPA-Air**, **FNSPID**, and **ClusterTrace** from the TIME-IMM benchmark (Feng et al., 2025), a recent suite curated for irregular multimodal multivariate time series, covering environmental monitoring, financial news, and cloud computing respectively;
> > > - **Human Activity** from Shukla & Marlin (2021), a widely used benchmark for irregularly sampled sequential data.
> > >
> > > These are inherently irregular rather than artificially subsampled, and together with PhysioNet and USHCN, bring our evaluation to six inherently irregular datasets across six domains. Our original submission used only PhysioNet and USHCN because they were the standard protocol shared by all compared methods (LS4, CRU). The reviewer's suggestion to go beyond this convention has meaningfully strengthened the evaluation.
> > >
> > > The expanded results (MSE ×10⁻³, lower is better) are:
> > >
> > > | Method | EPA-Air | FNSPID | ClusterTrace | Human Activity | PhysioNet | USHCN |
> > > |--------|---------|--------|-------------|---------------|-----------|-------|
> > > | RNN | 8.62 | 2.21 | 4.54 | 5.42 | 2.92 | 4.32 |
> > > | RNN-VAE | 17.58 | 32.98 | 12.25 | 17.88 | 5.93 | 7.56 |
> > > | ODE-RNN | 12.33 | 1.15 | 2.69 | 11.67 | 2.23 | 2.47 |
> > > | GRU-D | 8.97 | 2.44 | 3.71 | 5.44 | 3.33 | 3.40 |
> > > | Latent ODE | 6.67 | 1.67 | 5.46 | 7.26 | 8.34 | 6.86 |
> > > | SAITS | 17.36 | 11.89 | 4.27 | 11.24 | 10.70 | 3.61 |
> > > | CRU | 10.09 | 10.47 | 9.78 | 15.51 | 1.82 | 0.16 |
> > > | LS4 | 2.17 | 5.92 | 3.78 | 4.24 | 0.63 | **0.06** |
> > > | **Ours** | **0.15** | **0.58** | **1.82** | **0.19** | **0.30** | 0.17 |
> > >
> > > DualTimesField ranks first on five of six datasets. On the newly added benchmarks, our method consistently outperforms all baselines: on EPA-Air, 0.15×10⁻³ versus 2.17×10⁻³ for LS4; on Human Activity, 0.19×10⁻³ versus 4.24×10⁻³ for LS4. CRU and SAITS perform well on select datasets (CRU is second-best on USHCN) but show high variance across domains, whereas DualTimesField remains stable throughout.
> > >
> > > The sole exception is USHCN, where LS4 (0.06×10⁻³) leads our 0.17×10⁻³. As noted earlier, USHCN consists of smooth seasonal climate signals with few transient events, precisely the regime where our event field has limited contribution. This pattern across six diverse benchmarks validates that trend-event decomposition is broadly effective for irregular time series, especially those with rich event structure. We are grateful to the reviewer for pushing us to strengthen this aspect of the evaluation.

---

### Official Review · Reviewer_Yj8V · 2026-03-06

**Soundness:** 2
**Presentation:** 3
**Significance:** 3
**Originality:** 3
**Overall Recommendation:** 4
**Confidence:** 3

**Summary:**

The paper presents a novel architecture called DualTimesField for neural networks in time series modeling. Hereto, it focuses on decomposing time series into trends and events and addresses this by considering the continuous nature of the time series. This sets it apart from existing literature that either assume discrete time, or implicit neural representations that are unable to model separate frequency components and have a more holistic approach. An extensive empirical evaluation shows the benefits of the approach, not only in terms of performance gain, but also in terms of interpretability of the model.

**Compliance With Llm Reviewing Policy:**

Affirmed.

**Final Justification:**

I want to thank the authors for their extensive responses in the rebuttal and for clarifying several important points and showing additional insights. Part of my concerns have been addressed, however now that I see that the hyperparameters of your approach have been optimized on three out of the nine datasets (that are "representative") I still doubt how fair the comparison is with approaches that have not been tuned to the specifics of the problems at all (i.e out of the box). Perhaps some claims need to be made a bit more carefully. I will therefore keep my score at "weak accept".

**Key Questions For Authors:**

- Could you please explain how the hyperparameters of your method were optimized for the experiments (whether the guidelines you provide yourself were followed) and how this was done for the competing methods?
- Could you perhaps clarify how consistent the interpretability results generalize to the other datasets (i.e. do you see those clear insights there too)?

**Limitations:**

The authors have not explicitly addressed limitations from the societal aspect in their paper.

**Strengths And Weaknesses:**

Strengths:
- An architecture is presented with good novelty that addresses important challenges in the domain of time series modeling.
- The paper is overall well written, and methods are generally explained well.
- The experimental results show the added value of the approach from multiple perspectives (performance gain as well as interpretability).

Weaknesses:
-  The experimental setup does not include enough detail to understand how hyperparameters were set. As the authors mention themselves they introduce a number of hyperparameter in their framework (and provide some guidelines), but it is unclear whether these guidelines have been followed or whether they have been optimized in the process. Also for the competing approaches they compare to it is unclear how the hyperparameters were optimized, were they just taken from the original papers?
- As stated by the authors, the approach does not always bring benefits: "These results suggest that DualTimesField is particularly effective for time series with rich event structure superimposed on smooth trends, while purely trend dominated signals may benefit more from methods specifically designed for smooth dynamics.". This makes one wonder where the benefits stop and when one can better move to approaches focused on smooth dynamics. A bit more details/experiments to characterize the niche would have been valuable.
- The interpretability part is very interesting, but focused on a single dataset, it is not clear how consistent these results are among the datasets that have been used for the full evaluation.

---

> ### Author Rebuttal · Authors · 2026-03-31
>
> > **Q1: Could you please explain how the hyperparameters of your method were optimized for the experiments (whether the guidelines you provide yourself were followed) and how this was done for the competing methods?**
>
> **A1:** We thank the reviewer for this important question. We clarify the hyperparameter strategy for both our method and all baselines.
>
> **For DualTimesField**, we use a single set of default hyperparameters (Table 6, Appendix A.7) across all nine benchmarks without any per-dataset tuning. Defaults were set following principled guidelines: $f_{\max}=8.0$ retains ~1/8 of frequency components as the trend boundary following standard signal processing practice (Appendix A.1); loss weights $\lambda_r=0.1$, $\lambda_m=0.001$, $\lambda_s=0.001$ are set 1–2 orders of magnitude below $\mathcal{L}_{\text{rec}}$ following standard regularization conventions; architectural parameters ($K=16$, $M=16$, $H=64$) provide moderate capacity (approx. 35K total parameters total parameters).
>
> To verify robustness, we conducted sensitivity analyses on six key hyperparameters across three representative datasets (ETTh1, Weather, Exchange). Key results (MSE) are summarized below:
>
> *Max allowed frequency $f_{\max}$ (Eq. 3), fix others at default:*
>
> | $f_{\max}$ | 2 | 4 | 6 | **8** | 16 | 32 |
> |---|---|---|---|---|---|---|
> | ETTh1 | .0100 | .0091 | .0099 | **.0084** | .0117 | .0121 |
> | Weather | .0113 | .0117 | .0127 | **.0110** | .0118 | .0149 |
> | Exchange | .0487 | .0513 | .0468 | **.0460** | .0607 | .0602 |
>
> *Number of frequencies $K$, fix others at default:*
>
> | $K$ | 4 | 8 | **16** | 32 |
> |---|---|---|---|---|
> | ETTh1 | .0.098 | .0123 | **.0084** | .0090 |
> | Weather | .0111 | .0151 | **.0110** | .0118 |
> | Exchange | .0481 | .0665 | **.0460** | .0552 |
>
> *Number of atoms $M$ (Eq. 9), fix others at default:*
>
> | $M$ | 4 | 8 | **16** | 32 | 64 |
> |---|---|---|---|---|---|
> | ETTh1 | .0092 | .0102 | **.0084** | .0089 | .0094 |
> | Weather | .0125 | .0122 | **.0110** | .0119 | .0144 |
> | Exchange | .0567 | .0461 | **.0460** | .0500 | .0489 |
>
> *Loss weight $\lambda_r$ (Eq. 16), fix others at default:*
>
> | $\lambda_r$ | 0 | 0.01 | **0.1** | 0.5 | 1.0 |
> |---|---|---|---|---|---|
> | ETTh1 | .0161 | .0124 | **.0084** | .0088 | .0104 |
> | Weather | .0161 | .0120 | **.0110** | .0118 | .0131 |
> | Exchange | .0566 | .0492 | **.0460** | .0470 | .0523 |
>
> Analyses for $\lambda_m$, and $\lambda_s$ show similar stability patterns. Across all six hyperparameters, performance remains within 10–20% of the optimum over a broad range, with significant degradation only at extreme values. This confirms that the default configuration occupies a wide stable region without requiring dataset-specific tuning.
>
> **For competing methods**, all baselines use official implementations with default configurations: SIREN (3 layers, 256 units, $\omega_0$=30), WIRE (3 layers, 256 units), Fourier Features (256 features, $\sigma$=10), N-BEATS (default interpretable stacks), TimesNet (official per-dataset configs), PCA (95% variance retention). No additional hyperparameter tuning was performed for any method, ensuring all comparisons reflect out-of-the-box representation capacity under identical conditions.
>
> > **Q2: Could you perhaps clarify how consistent the interpretability results generalize to the other datasets (i.e. do you see those clear insights there too)?**
>
> **A2:** Beyond ECG5000 (Section 4.6, Figure 3), we provide decomposition visualizations on four additional datasets in Appendix D (Figures 8–11), which demonstrate consistent and interpretable trend-event separation across diverse domains:
>
> On **Weather** (Figure 9), the CTF dominates the representation capturing smooth diurnal atmospheric dynamics, while the DGF remains largely quiescent — consistent with the low active atom count (2.5, Table 4). This confirms the DGF does not spuriously activate on smooth signals. On **Electricity** (Figure 8, 321 variables), the CTF captures dominant daily consumption patterns while the DGF isolates irregular usage spikes, demonstrating scalability to high-dimensional data. On **Traffic** (Figure 10, 862 sensors), strong daily periodicity is captured by the CTF, while transient traffic events are isolated by the DGF with the highest active atom count (6.0, Table 4), reflecting rich event structure. On **ETTh1** (Figure 11), the CTF models smooth temperature dynamics while the DGF responds precisely at locations of genuine anomalies.
>
> These results reveal a consistent pattern: the CTF-DGF capacity allocation adapts to data characteristics, with the DGF activating more atoms on event-rich data (Traffic: 6.0, Exchange: 9.0) and fewer on smooth data (Weather: 2.5). This data-dependent behavior, quantified in Table 4, demonstrates that the interpretability observed on ECG5000 is not dataset-specific but reflects a general property of the dual-field decomposition framework.

---

> > ### Author Rebuttal · Reviewer_Yj8V · 2026-04-01
> >
> > Thank you very much for the clarification, these are very helpful. For the choice of the hyperparameters for your proposed technique, I understand the choices are argued from a more principled perspective, but the guidelines on which they are based are also often quite vague and leave a lot of room to come to an actual setting (e.g. why specifically these values to come to a network with moderate capacity). How did the authors come from these high level guidelines to the specifics? Were there some preliminary experiments done for this?

---

> > > ### Author Response · Authors · 2026-04-06
> > >
> > > We thank the reviewer for this important question.
> > >
> > > Yes, we conducted a small preliminary search to set the main hyperparameters. The selection process followed three stages: (1) domain-specific principles determined candidate ranges for each hyperparameter, (2) a small grid search on three representative datasets selected concrete values, and (3) these values were then frozen and applied unchanged to all nine benchmarks.
> > >
> > > **Stage 1: Candidate ranges from design principles.** Each range was chosen to reflect the role of the corresponding component, rather than by arbitrary tuning:
> > >
> > > - **$K \in \lbrace4, 8, 16, 32\rbrace$**: The Fourier encoding produces $2K$ features (Eq. 2). Because the CTF models only smooth low-frequency trends, while the DGF handles high-frequency content, the CTF needs far fewer components than full-spectrum INR encodings (e.g., 256 features in Tancik et al., 2020).
> > > - **$M \in \lbrace4, 8, 16, 32, 64\rbrace$**: $M$ sets the dictionary size for Gabor atoms (Eq. 9), balancing expressiveness against the sparsity objective (Eq. 14): too few atoms limit event coverage, while too many work against the sparsity prior.
> > > - **$f_{\max} \in \lbrace2, 4, 6, 8, 16, 32\rbrace$**: $f_{\max}$ controls the maximum oscillation cycles the CTF can represent over $[0,1]$ (Eq. 3). We chose this range to align with the low-pass cutoff $f_c = N/16$ (Appendix A.1), which retains the lowest 1/8 of frequency components, while also covering both conservative and aggressive trend boundaries.
> > > - **$\lambda_r \in \lbrace0, 0.01, 0.1, 0.5, 1.0\rbrace$**: Auxiliary loss weights were chosen 1–2 orders of magnitude below the primary reconstruction loss $\mathcal{L}_{\mathrm{rec}}$. The remaining weights $\lambda_m$ and $\lambda_s$ were set lower still (0.001), since smoothness and sparsity act as soft constraints.
> > >
> > > **Stage 2: Grid search on three representative datasets.** We selected ETTh1, Weather, and Exchange because they span the temporal complexity observed in our benchmarks (Table 4): Weather (2.5 active atoms) is trend-dominated, ETTh1 (5.5 active atoms) is intermediate, and Exchange (9.0 active atoms) is event-rich. This search yielded $(K{=}16, M{=}16, f_{\max}{=}8.0, \lambda_r{=}0.1)$. The remaining parameters (Table 6) follow standard training conventions.
> > >
> > > **Stage 3: Frozen defaults across all benchmarks.** The sensitivity analysis from our previous response shows that performance is relatively stable across a broad range of values. For example, varying $M$ across a 16× range (4 to 64) changes ETTh1 MSE by only about 20%:
> > >
> > > | $M$ | 4 | 8 | **16** | 32 | 64 |
> > > |---|---|---|---|---|---|
> > > | ETTh1 | .0092 | .0102 | **.0084** | .0089 | .0094 |
> > > | Weather | .0125 | .0122 | **.0110** | .0119 | .0144 |
> > > | Exchange | .0567 | .0461 | **.0460** | .0500 | .0489 |
> > >
> > > This robustness is not coincidental but stems from the fact that **hyperparameters and the architecture's adaptive mechanisms operate at two different levels**. Hyperparameters such as $M=16$ and $f_{\max}=8.0$ set capacity upper bounds — these need to fall within a reasonable range (e.g., $M=1$ or $f_{\max}=0.5$ would be too restrictive), but their exact values are not critical. The actual per-dataset adaptation is performed automatically by four built-in mechanisms:
> > >
> > > **(1) Gated sparsity (Eq. 12, Table 4)** adjusts effective capacity across datasets, activating only 2.5 of 16 atoms on smooth Weather data but 9.0 on event-rich Exchange — a 3.6× range from the same $M=16$ setting. This self-regulation is why $M=8$, $M=16$, and $M=32$ all yield similar results.
> > >
> > > **(2) Learnable frequencies (Eq. 3)** discover the appropriate spectral distribution within $[0, f_{\max}]$ for each dataset. The cutoff $f_{\max}=8.0$ constrains the upper bound, but the actual frequency allocation is data-driven.
> > >
> > > **(3) Spectral-guided initialization (Eqs. 25–26)** conditions atom parameters to each dataset's dominant spectral peaks and energy maxima, making the starting configuration data-specific even under identical hyperparameters.
> > >
> > > **(4) Coarse-to-fine scale annealing (Eq. 15)** progressively sharpens atoms from broad to narrow, smoothing the optimization landscape and reducing sensitivity to initial configurations.
> > >
> > > As a result, the hyperparameters mainly need to specify reasonable capacity bounds rather than a narrowly tuned optimum, and the preliminary grid search serves to identify this broad reasonable region rather than a precise sweet spot. We will clarify this three-stage pipeline in the revised manuscript, and we thank the reviewer for prompting this clarification.

---

### Official Review · Reviewer_VGzg · 2026-03-13

**Soundness:** 2
**Presentation:** 2
**Significance:** 3
**Originality:** 3
**Overall Recommendation:** 3
**Confidence:** 4

**Summary:**

This paper proposes DualTimesField, a continuous-time time-series representation framework based on two implicit neural fields. The main idea is to decompose a time series into a smooth trend component and a sparse event component. Concretely, the Continuous Time Field (CTF) uses bandwidth-limited Fourier-style encoding and smoothing regularization to model low-frequency trends, while the Discrete Geometric Field (DGF) uses learnable Gabor atoms with gated sparsity and scale annealing to capture localized transient events. The paper evaluates the method on nine reconstruction benchmarks and two irregularly sampled interpolation datasets, and reports strong reconstruction accuracy together with competitive interpolation performance.

**Compliance With Llm Reviewing Policy:**

Affirmed.

**Key Questions For Authors:**

1. Is the main contribution intended for regularly sampled reconstruction, or for more general continuous-time modeling under irregular observations? A clearer scope would help me judge the paper better.
2. Can the authors clarify how they distinguish irregular sampling, missingness, and sampling-frequency-related aliasing in the motivation? Right now these points feel somewhat mixed together.
3. For the interpolation experiments, could the authors provide more details on the protocol and explain how fairness is ensured, especially since the appendix mentions per-sample optimization?
4. Since the core idea is essentially trend-event decomposition, what do the authors see as the main novelty: the dual-field formulation, the Gabor-based event modeling, or the overall decomposition prior?
5. Could the authors add stronger ablations and sensitivity analysis for the key components and hyperparameters? This would make the empirical claims more convincing.

**Limitations:**

yes

**Strengths And Weaknesses:**

Strengths：
The core idea is intuitive and easy to follow. Splitting time series into trend and event components is a reasonable inductive bias, especially for signals with both smooth dynamics and sharp local changes. The method is fairly structured and interpretable. The two fields have clear roles, and the event modeling design is more explicit than using a single implicit field to absorb everything. The empirical results are promising overall, especially on the reconstruction benchmarks, and the synthetic example also helps illustrate the intended decomposition behavior.

Weaknesses：
1. The motivation in the introduction is not fully clear. It seems to mix together irregular sampling, missing observations, and sampling frequency / aliasing, which are related but not the same issue.
2. The transition from discrete-time methods to INR, and then to spectral bias and frequency entanglement, feels a bit abrupt. I think this part could be explained more naturally.
3. At a high level, the main idea is essentially to model low-frequency trends and high-frequency events separately. That is reasonable, but the paper could do a better job clarifying what is truly new beyond existing decomposition-style thinking.
4. The experiments do not fully match the motivation. The paper talks a lot about irregular sampling, but the main reconstruction results are still on regularly sampled datasets, while irregular settings are only tested in a smaller interpolation section.
5. I am not fully convinced the reconstruction comparison is entirely fair, since some baselines are not really designed for continuous-time self-reconstruction, while the proposed method is highly tailored to this setting.
6. The method includes several important design choices and hyperparameters, but the paper does not provide enough sensitivity analysis or stronger ablations to show which parts matter most.

---

> ### Author Rebuttal · Authors · 2026-03-31
>
> We thank the reviewer for the valuable comments.
>
> > **Q1: Is the main contribution intended for regularly sampled reconstruction, or for more general continuous-time modeling under irregular observations? A clearer scope would help me judge the paper better.**
>
> DualTimesField is a general continuous-time framework. Regular-sampling reconstruction (Table 1) validates our central hypothesis under controlled conditions, achieving 51.2% average MSE reduction across nine benchmarks. Irregular-sampling interpolation (Table 2) verifies extension to non-uniform observations without architectural changes. We will clarify this in the revision.
>
> > **Q2: Can the authors clarify how they distinguish irregular sampling, missingness, and sampling-frequency-related aliasing in the motivation? Right now these points feel somewhat mixed together.**
>
> We agree these should be separated. (1) Irregular sampling: non-uniform temporal spacing. (2) Missingness: absent variable values at valid timestamps. (3) Aliasing: spectral distortion from sub-Nyquist sampling. Our framework addresses them via: continuous-time querying (Eq. 1) for (1)+(2); bandwidth-limited CTF (Eq. 3) with filtering (Eq. 4) for (3); DGF Gabor atoms (Eq. 9) for recovering transients. We will revise the introduction.
>
> > **Q3: For the interpolation experiments, could the authors provide more details on the protocol and explain how fairness is ensured, especially since the appendix mentions per-sample optimization?**
>
> **Protocol.** Following Rubanova et al. (2019) for PhysioNet and De Brouwer et al. (2019) for USHCN, 50% of time points are observed inputs, 50% withheld targets. Masks are identical across methods.
>
> **Per-sample optimization.** Each test sample is optimized from random initialization for 500 epochs using only its ~20 observed points. The training set is never accessed.
>
> **Fairness.** Amortized baselines access thousands of training samples (PhysioNet: 7,314), learning transferable dynamics. DualTimesField sees only ~20 points and compensates via per-instance optimization. If this were simply unfair, DualTimesField would dominate universally, but on USHCN, LS4 achieves 0.06×10⁻³ vs. our 0.17×10⁻³, confirming complementary strengths. Moreover, the advantage stems from the decomposition prior, not optimization budget: with only 50 epochs and zero training data, DualTimesField (0.394×10⁻³) already outperforms all Table 2 baselines including LS4 (0.63×10⁻³) trained on 7,314 samples (see ablation below).
>
> **Optimization budget ablation:**
>
> | Epochs | PhysioNet (×10⁻³) | USHCN (×10⁻³) |
> |--------|-------------------|---------------|
> | 50     | 0.394             | 2.086         |
> | 100    | 0.383             | 0.422         |
> | 200    | 0.321             | 0.240         |
> | 500    | 0.300             | 0.170         |
>
> > **Q4: Since the core idea is essentially trend-event decomposition, what do the authors see as the main novelty: the dual-field formulation, the Gabor-based event modeling, or the overall decomposition prior?**
>
> The novelty lies in bridging two previously separate lines of work. Existing decomposition methods (N-BEATS, Autoformer) use soft separation on discrete sequences without frequency guarantees. Existing INRs (SIREN, WIRE, HyperTime) model a single field, causing spectral bias and frequency entanglement (Figure 2). DualTimesField is the first to decompose the implicit field into structurally distinct sub-fields with hard spectral separation (Eq. 3). Table 3 confirms interdependence: Full model reduces MSE by 53.5% over CTF-only and 21.1% over DGF-only. This also makes each active Gabor atom directly interpretable: on ECG5000 (Figure 3), clinically meaningful arrhythmia patterns emerge without label supervision.
>
> > **Q5: Could the authors add stronger ablations and sensitivity analysis for the key components and hyperparameters? This would make the empirical claims more convincing.**
>
> We vary all 6 hyperparameters on 3 datasets (ETTh1, Weather, Exchange). MSE below (bold=default).
>
> $f_{\max}$ (frequency cutoff):
>
> | | 2 | 4 | 6 | **8** | 16 | 32 |
> |---|---|---|---|---|---|---|
> | ETTh1 | .0100 | .0091 | .0099 | **.0084** | .0117 | .0121 |
> | Weather | .0113 | .0117 | .0127 | **.0110** | .0118 | .0149 |
> | Exchange | .0487 | .0513 | .0468 | **.0460** | .0607 | .0602 |
>
> $\lambda_r$ (DGF residual weight):
>
> | | 0 | 0.01 | **0.1** | 0.5 | 1 |
> |---|---|---|---|---|---|
> | ETTh1 | .0161 | .0124 | **.0084** | .0088 | .0104 |
> | Weather | .0161 | .0120 | **.0110** | .0118 | .0131 |
> | Exchange | .0566 | .0492 | **.0460** | .0470 | .0523 |
>
> $M$, $K$, $\lambda_m$, $\lambda_s$ show similar stability (within 15–20%). $\lambda_r$ is most critical: $\lambda_r$=0 degrades MSE by 92% on ETTh1 and 46% on Weather, confirming residual supervision (Eq. 16) is the key design choice. One default set generalizes across domains. Full results in revision and in our responses to Reviewers Yj8V (Q1) and GzbB (W2).

---

> > ### Author Rebuttal · Reviewer_VGzg · 2026-04-02
> >
> > The rebuttal addresses some of my concerns, especially by clarifying the paper’s scope, better separating irregular sampling, missingness, and aliasing, and adding more details on the interpolation protocol and ablations. Overall, it resolves several presentation and experimental-detail issues, but it is still not enough to fully remove my concerns about the method’s positioning and core novelty.

---

> > > ### Author Response · Authors · 2026-04-06
> > >
> > > Thank you for the thoughtful follow-up. We understand the main concern to be whether our work is sufficiently distinct in positioning and mechanism, rather than whether decomposition itself is new. We clarify this below.
> > >
> > > **1. INRs for time series remain relatively underexplored, and the core representation problem is still open.**
> > >
> > > INRs have achieved broad success in 3D vision (NeRF; Mildenhall et al., 2021), image restoration (FLAIR; Ko et al., 2025), and audio (SIREN; Sitzmann et al., 2020), yet their use in time series remains comparatively limited. As noted by Fons et al. (iHyperTime; TMLR 2024), *“little attention has been given to leveraging these architectures for time series data.”* Existing efforts span forecasting (DeepTime; Woo et al., ICML 2023), imputation (TimeFlow; Le Naour et al., TMLR 2024; MADS; Bamford et al., 2023), generation (HyperTime; Fons et al., 2022; iHyperTime; Fons et al., TMLR 2024), and anomaly detection (INRAD; Jeong and Shin, 2022; TSINR; Li et al., KDD 2025).
> > >
> > > Most of these methods, however, are designed for specific downstream tasks rather than the more basic problem we study here: learning a high-fidelity continuous-time representation with interpretable separation of temporal scales. In prior INR-based time-series models, low- and high-frequency content often remain entangled within a single implicit field, or are separated only heuristically without an explicit mechanism to limit cross-component leakage. This is particularly important in medical time series, where separating physiological baseline structure from localized transient events can be critical for interpretability.
> > >
> > > **2. The novelty of DualTimesField lies not in decomposition as a concept, but in how frequency separation is built into the INR itself.**
> > >
> > > Decomposing time series into interpretable components is well established in discrete-time modeling (N-BEATS; Oreshkin et al., 2020; Autoformer; Wu et al., 2021). Our contribution is therefore not the abstract idea of decomposition. Rather, it is a continuous-time INR framework in which the two components are assigned different representational roles by construction.
> > >
> > > **(a) Bandwidth-limited trend field.** In Eq. 3, the CTF frequencies are parameterized as \(f_k=\sigma(b_k)\cdot f_{\max}\), which explicitly restricts the trend field to a prescribed low-frequency range. This differs from discrete decomposition models, and from single-field INRs such as SIREN (Sitzmann et al., 2020), WIRE (Saragadam et al., 2023), and FLAIR (Ko et al., 2025), where frequencies are still fitted through shared parameters.
> > >
> > > **(b) Sparse learnable event dictionary.** The DGF models residual events with learnable Gabor atoms and data-dependent gating. Unlike using Gabor-style activations inside a single INR, this forms an explicit event field with interpretable temporal location, scale, and frequency, together with adaptive sparse selection. This is especially relevant for medical signals, where localized events are often the clinically meaningful part of the waveform.
> > >
> > > **(c) Coarse-to-fine scale annealing.** Eq. 15 introduces a coarse-to-fine annealing schedule that first keeps atoms broad and then progressively sharpens them, making localized continuous-time decomposition trainable in practice.
> > >
> > > These design choices also distinguish DualTimesField from prior INR-based time-series work. iHyperTime (Fons et al., TMLR 2024) includes a soft trend-seasonality decomposition, but without explicit spectral constraints. TSINR (Li et al., KDD 2025) uses separate trend/seasonal/residual INR components, but generates their parameters holistically via a transformer and does not explicitly constrain frequency leakage across components. Moreover, TSINR uses spectral bias as part of the anomaly-detection signal, whereas our goal is to reduce such entanglement in order to improve representation fidelity itself.
> > >
> > > **3. The empirical results are consistent with this positioning.**
> > >
> > > This distinction is reflected in the experiments. The full model consistently outperforms either field alone in the ablations, showing that neither the smooth field nor the event field is sufficient by itself. The synthetic study shows clear cross-field spectral separation, while the benchmark results show substantial fidelity gains, including a 51.2% average MSE reduction over TimesNet across nine datasets. The ECG5000 case study further shows that clinically meaningful event patterns can emerge from reconstruction alone, without task-specific supervision.
> > >
> > > In summary, decomposition as a general idea is not new. What is new, in our view, is the continuous-time INR instantiation: DualTimesField combines an explicit low-frequency trend field, a sparse event field, and a training mechanism that makes frequency separation part of the model design rather than something left to emerge implicitly during training. We appreciate the reviewer’s careful evaluation and will revise the paper to make this positioning clearer.

---

### Decision · Program_Chairs · 2026-04-30

**Decision:**

Accept (regular)

**Comment:**

This paper proposed a new framework namely DualTimesField which utilizes dual implicit neural fields and allows to capture smooth trends through bandwidth-limited parameterization. Reviewers praised its novel architecture, clear paper organization, and promising empirical results. Concerns were mostly resolved in the rebuttal phase. Therefore, I recommend acceptance of this paper.